# Tree islands enhance biodiversity and functioning in oil palm landscapes

Delphine Clara Zemp[1,2,3 ✉], Nathaly Guerrero-Ramirez[2], Fabian Brambach[2], Kevin Darras[4], Ingo Grass[5], Anton Potapov[6], Alexander Röll[7], Isabelle Arimond[4,8], Johannes Ballauff[9], Hermann Behling[10], Dirk Berkelmann[11], Siria Biagioni[12], Damayanti Buchori[13,14], Dylan Craven[15], Rolf Daniel[11], Oliver Gailing[3,16], Florian Ellsäßer[7,17], Riko Fardiansah[3,18,19], Nina Hennings[20], Bambang Irawan[19], Watit Khokthong[7,21], Valentyna Krashevska[6], Alena Krause[6], Johanna Kückes[7], Kevin Li[4], Hendrik Lorenz[7], Mark Maraun[6], Miryam Sarah Merk[22], Carina C. M. Moura[16], Yeni A. Mulyani[23], Gustavo B. Paterno[2], Herni Dwinta Pebrianti[19], Andrea Polle[3,9], Di Ajeng Prameswari[6], Lena Sachsenmaier[2,24,25], Stefan Scheu[3,6], Dominik Schneider[11], Fitta Setiajiati[26], Christina Ani Setyaningsih[10], Leti Sundawati[26], Teja Tscharntke[4], Meike Wollni[3,27], Dirk Hölscher[3,7] & Holger Kreft[2,3]

In the United Nations Decade on Ecosystem Restoration[1], large knowledge gaps persist on how to increase biodiversity and ecosystem functioning in cash crop-dominated tropical landscapes[2]. Here, we present findings from a large-scale, 5-year ecosystem restoration experiment in an oil palm landscape enriched with 52 tree islands, encompassing assessments of ten indicators of biodiversity and 19 indicators of ecosystem functioning. Overall, indicators of biodiversity and ecosystem functioning, as well as multidiversity and ecosystem multifunctionality, were higher in tree islands compared to conventionally managed oil palm. Larger tree islands led to larger gains in multidiversity through changes in vegetation structure. Furthermore, tree enrichment did not decrease landscape-scale oil palm yield. Our results demonstrate that enriching oil palm-dominated landscapes with tree islands is a promising ecological restoration strategy, yet should not replace the protection of remaining forests.

The loss of megadiverse tropical lowland rainforests has accelerated in the past decades[3], with deforestation and land-use change being largely driven by the rapid expansion of high-yielding cash crops such as oil palm[4]. Globally, oil palm plantations occupy 21 million hectares, mostly in Indonesia and Malaysia[5]. Although the expansion of oil palm has promoted economic development and improved livelihoods of smallholder farmers, it has also led to dramatic negative ecological impacts[6]. Compared with tropical lowland rainforests, species diversity in oil palm-dominated landscapes is greatly reduced[7], especially for forest-dependent species and species of conservation concern[4]. In addition, the transformation of forests to oil palm-dominated landscapes alters the functioning of ecological communities and environmental conditions, leading to a reduction of several ecosystem functions and services[7,8].

Many agricultural landscapes are in urgent need of ecological restoration to safeguard biodiversity and ecosystem functioning while also promoting local livelihoods[9–11], a central goal of the current United Nations decade on Ecosystem Restoration. However, trade-offs between biodiversity or ecosystem functioning and agricultural productivity may result in failed restoration efforts or lead to undesirable ecological spillover effects by promoting the expansion of the agricultural frontier into natural forested areas[12]. One way to mitigate

[1]Conservation Biology, Institute of Biology, Faculty of Sciences, University of Neuchâtel, Neuchâtel, Switzerland. [2]Biodiversity, Macroecology and Biogeography, Faculty of Forest Sciences and Forest Ecology, University of Göttingen, Göttingen, Germany. [3]Centre of Biodiversity and Sustainable Land Use (CBL), University of Göttingen, Göttingen, Germany. [4]Agroecology, Department of Crop Sciences, Faculty of Agricultural Science, University of Göttingen, Göttingen, Germany. [5]Ecology of Tropical Agricultural Systems, Institute of Agricultural Sciences in the Tropics, University of Hohenheim, Stuttgart, Germany. [6]Animal Ecology, J.F. Blumenbach Institute of Zoology and Anthropology, University of Göttingen, Göttingen, Germany. [7]Tropical Silviculture and Forest Ecology, Faculty of Forest Sciences and Forest Ecology, University of Göttingen, Göttingen, Germany. [8]Functional Agrobiodiversity, Dept. of Crop Sciences, Faculty of Agricultural Science, University of Göttingen, Göttingen, Germany. [9]Forest Botany and Tree Physiology, Faculty of Forest Sciences and Forest Ecology, University of Göttingen, Göttingen, Germany. [10]Department of Palynology and Climate Dynamics, Albrecht-von-Haller-Institute for Plant Sciences, University of Göttingen, Göttingen, Germany. [11]Department of Genomic and Applied Microbiology, Institute of Microbiology and Genetics, University of Göttingen, Göttingen, Germany. [12]Department of Palynology and Climate Dynamics, Albrecht-von-Haller-Institute for Plant Sciences, University of Göttingen, Göttingen, Germany. [13]Department of Plant Protection, Faculty of Agriculture, Institut Pertanian Bogor. Jl. Meranti, IPB Dramaga Campus, Bogor, Indonesia. [14]Center for Transdisciplinary and Sustainability Sciences, IPB University, Jalan Pajajaran, Indonesia. [15]Centre for Ecosystem Modeling and Monitoring, Facultad de Ciencias, Universidad Mayor, Santiago, Chile. [16]Forest Genetics and Forest Tree Breeding, Faculty of Forest Sciences and Forest Ecology, University of Göttingen, Göttingen, Germany. [17]Department of Natural Resources, University of Twente, Enschede, Netherlands. [18]Zoological Museum, Center of Natural History, Universität Hamburg, Hamburg, Germany. [19]Faculty of Forestry, University of Jambi Jln Raya Jambi, Jambi, Indonesia. [20]Biogeochemistry of Agroecosystems, Faculty of Agricultural Science, University of Göttingen, Göttingen, Germany. [21]Department of Biology, Faculty of Science, Chiang Mai University, Chiang Mai, Thailand. [22]Chairs of Statistics and Econometrics, Faculty of Business and Economics, University of Göttingen, Göttingen, Germany. [23]Forest Resources Conservation and Ecotourism, Faculty of Forestry and Environment, IPB University, Kampus IPB Darmaga, Bogor, Indonesia. [24]German Centre for Integrative Biodiversity Research (iDiv) Halle-Jena-Leipzig, Leipzig, Germany. [25]Systematic Botany and Functional Biodiversity, Institute of Biology, Leipzig University, Leipzig, Germany. [26]Department of Forest Management, Faculty of Forestry and Environment, IPB University, Kampus IPB Darmaga, Bogor, Indonesia. [27]Environmental and Resource Economics, Department of Agricultural Economics and Rural Development, Faculty of Agricultural Sciences, University of Göttingen, Göttingen, Germany. ✉e-mail: clara.zemp@unine.ch

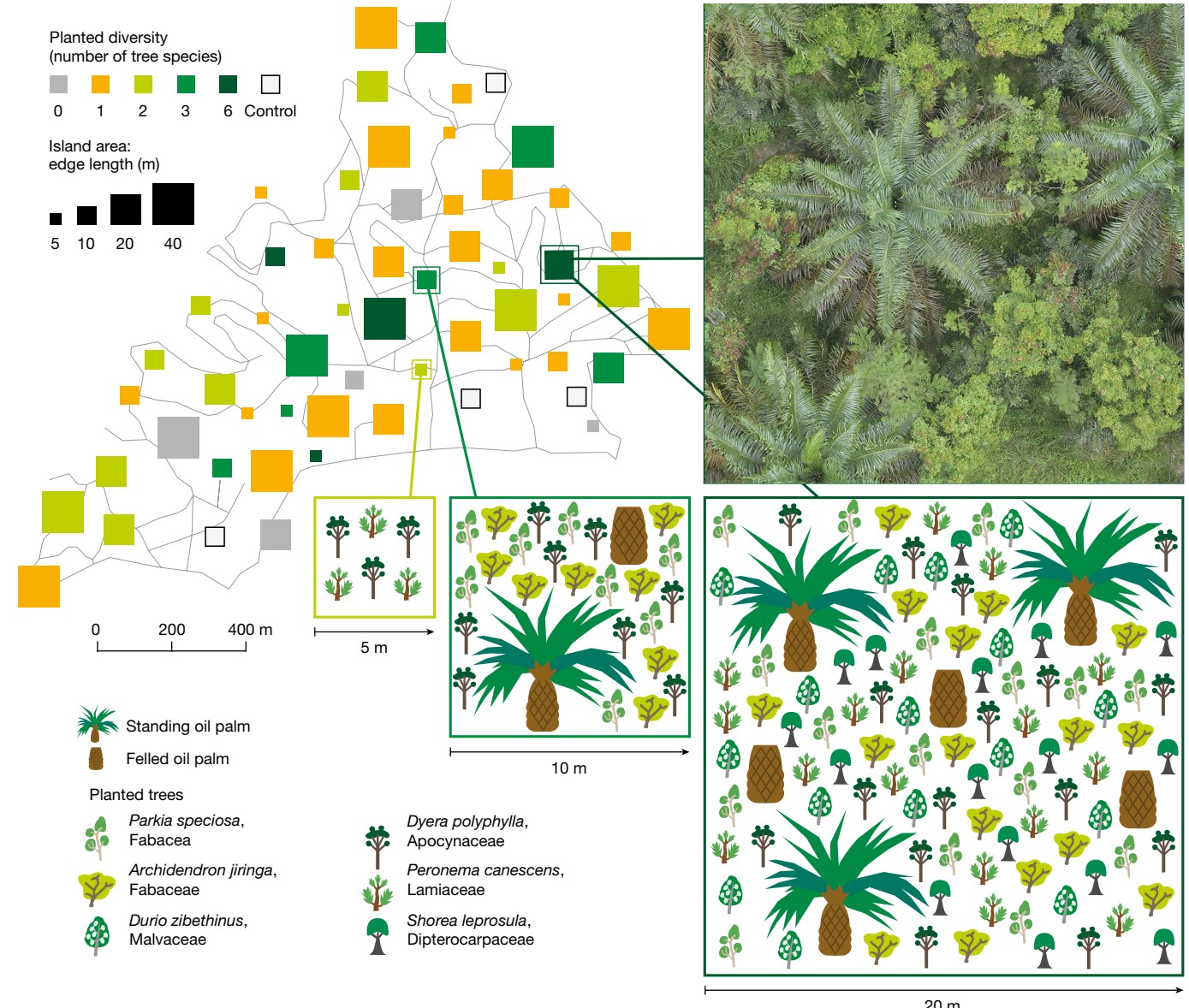

**Fig. 1 | Experimental design that tests the ecological restoration outcomes of tree island establishment in oil palm-dominated landscapes.** Tree islands vary in area (25–1,600 m²) and planted tree diversity (none to six species), with a total of 52 tree islands established in an industrial oil palm plantation in Sumatra, Indonesia. Control plots represent conventionally managed oil palm monocultures. Note that the islands in the map are not at scale.

trade-offs between restoration outcomes is to enrich agricultural landscapes with species-rich agroforestry systems[13,14] and islands of native trees through planting or natural regeneration[15–17]. However, to be a viable alternative for landowners, it is essential to generate empirical evidence on whether and how these restoration strategies affect biodiversity, ecosystem functioning and agricultural productivity in cash crop-dominated landscapes[2].

Here, we present the results of a large-scale, interdisciplinary ecosystem restoration experiment, in which the restoration outcomes across 52 tree islands established in a landscape dominated by an industrial oil palm plantation (140 ha) were observed and quantified three to five years after establishment. We assessed above- and below-ground biodiversity across ten indicators representing a broad range of Kingdoms (bacteria, fungi, plants and animals; Supplementary Table 1) and 19 indicators of ecosystem functioning associated with productivity of oil palms and planted trees, resistance to invasion, pollination, soil quality, predation and herbivory, carbon and nutrient cycling and water and climate regulation (Supplementary Table 2). To provide an holistic

overview of biodiversity and ecosystem functioning across the experiment, we calculated multidiversity and multifunctionality using the aforementioned indicators[18]. The experimental design allowed us to test the effects of tree island area (25, 100, 400 and 1,600 m²) and of planted native tree diversity (zero, one, two, three and six species, with zero representing natural regeneration only) on restoration outcomes and to compare them with conventionally managed oil palm monocultures[19] (Fig. 1). Overall, we expected tree islands to enhance biodiversity and ecosystem functioning compared to conventionally managed oil palm monocultures. To provide a mechanistic understanding of the effects of planted tree diversity and island area on biodiversity and ecosystem functioning, we also measured 12 indicators of vegetation structure (Supplementary Table 3). On the basis of the theory of island biogeography[20], we expected larger tree islands to have enhanced biodiversity and ecosystem functioning compared to smaller ones. Larger tree islands potentially provide more habitats and sustain larger populations, whereas smaller islands are expected to be more like the surrounding environment, that is, the oil palm-dominated landscape.

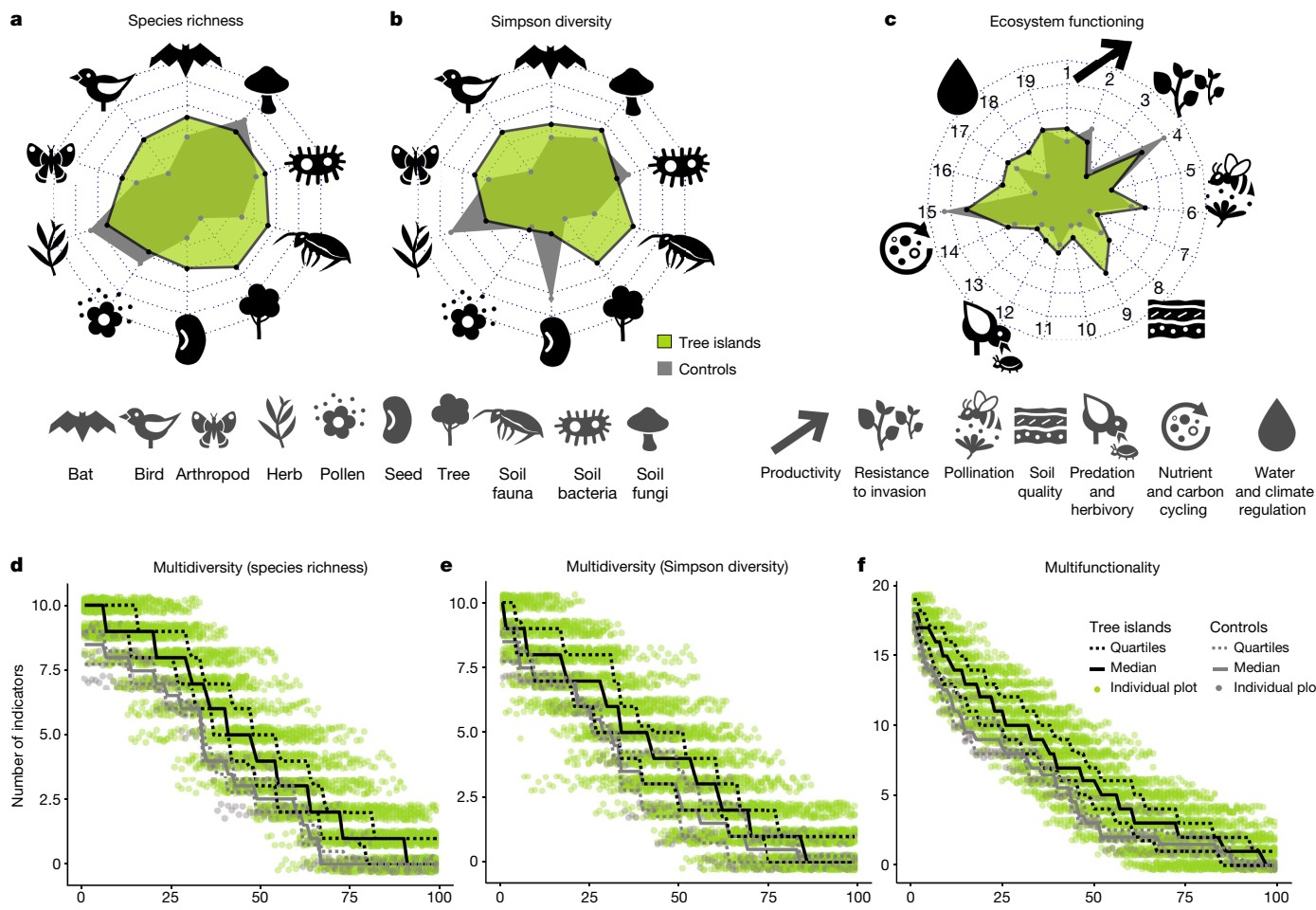

**Fig. 2 | Multidimensional ecological restoration outcomes in an oil palm-dominated landscape.** We measured 10 and 19 indicators of biodiversity and ecosystem functioning, respectively, in tree islands and compared their responses to those in plots representing conventionally managed oil palm monocultures. For ecosystem functioning, we measured: productivity as (1) oil palm yield and (2) above-ground biomass; resistance to invasion of (3) native seeds and (4) resistance to invasive plants; pollination as (5) pollinators and (6) pollination rate; soil quality as (7) soil P, (8) soil decompaction and (9) 1/soil C:N; predation and herbivory as (10) predators (vertebrates), (11) predators (arthropods), (12) predators (soil fauna) and (13) herbivores (soil fauna); carbon and nutrient cycling as (14) decomposers, (15) litter decomposition and (16) litter input; water and climate regulation as (17) evapotranspiration, (18) water infiltration and (19) microclimate buffering. Oil palm yield (calculated per island) is considered as an ecosystem functioning because of its contribution to primary productivity, as well as agricultural productivity. **a–c**, Indicators of biodiversity calculated as species richness (**a**) and Simpson diversity (**b**), which emphasizes the contribution of abundant species and ecosystem functioning (**c**) across 52 tree islands (green polygons) compared to four control plots of conventionally managed oil palm monocultures (grey polygons). **d–f**, Polygon vertices represent median values for each indicator. Multidiversity and multifunctionality represent the number of indicators (species richness (**d**); Simpson diversity (**e**) and ecosystem functioning (**f**)) that exceed a specified threshold, which is expressed as a percentage of the maximum observed values in the oil-dominated landscape (calculated on the basis of both island and control plots combined).

We further expected greater planted tree species diversity to favour diversity at higher trophic levels[21] and enhance ecosystem functioning through complementarity among species[22]. Planted diversity effects on restoration outcomes are probably mediated by higher vegetation structural complexity, that is, the three-dimensional distribution of plants within an ecosystem[23]. Finally, we proposed that agricultural productivity (oil palm yield) decreases at the local scale (within tree islands), whereas the loss is negligible at the scale of the industrial plantation or landscape[16].

Tree islands had higher biodiversity and ecosystem functioning compared to conventionally managed oil palm monocultures (Fig. 2 and Extended Data Table 1). Yet, tree island effects on biodiversity varied depending on the indicator (tree island × indicator: $F = 2.5$, $P = 0.007$ for species richness; $F = 3.6$, $P$ 0.0002 for Shannon diversity; and $F = 3.0$, $P = 0.001$ for Simpson diversity; Extended Data Table 1 and Extended Data Fig. 1). For example, natural regeneration and colonization led to increases in tree and bird species richness (+4.7 tree species in islands compared to monocultures, +2.5 bird species) and decreases in the diversity of the most abundant seed species (−1.2 seed species based on Simpson diversity; Supplementary Table 4). Overall, restoration benefits of tree islands were found for ecosystem functioning (tree island: $F = 6.2$, $P = 0.016$; Extended Data Table 1); with strongest increases for water infiltration (+174% saturated soil hydraulic conductivity), litter input (+151% leaf litter biomass input), activity of insectivorous bats and birds (+556%) and soil fertility (+14% 1/soil C:N ratio; Supplementary Table 5). Overall, multidiversity and ecosystem multifunctionality were higher in tree islands than in conventionally managed oil palm monocultures, regardless of the threshold used for calculation or when considering relative species abundances (Fig. 2d–f). The calculation of multidiversity (or multifunctionality) relies on the number of biodiversity (or functioning) indicators that cross a certain threshold, with thresholds expressed as the percentage of the maximum observed

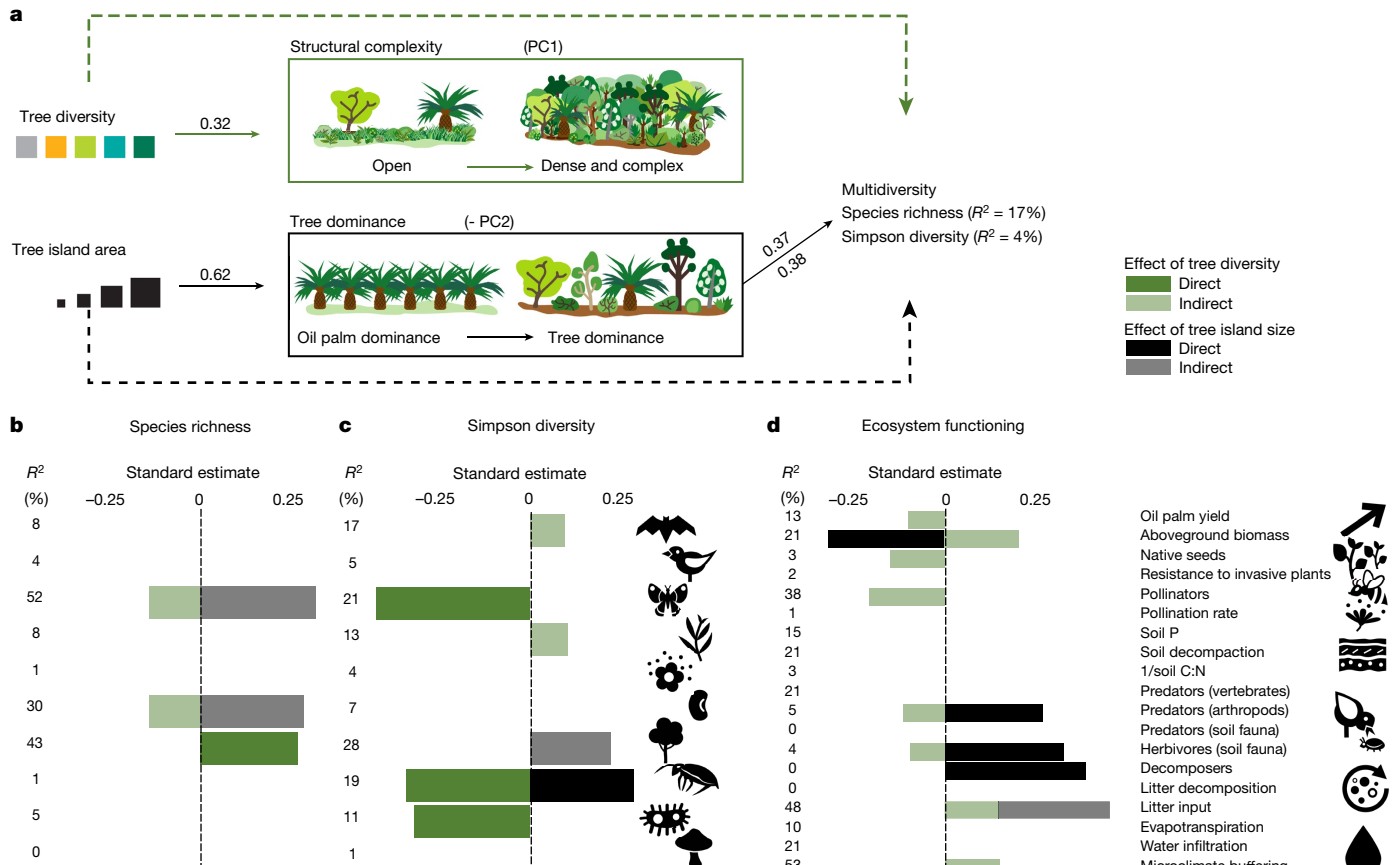

**Fig. 3 | Influence of tree island area and planted tree diversity on multidimensional restoration outcomes in an oil palm-dominated landscape. a**, Effects of planted tree diversity (directly or through structural complexity) and tree island area (directly or through changes in tree dominance) on multidiversity and multifunctionality tested with SEMs. Filled arrows (and standard coefficient estimates) indicate statistically significant effects ($P < 0.05$; two-sided analysis of variance (ANOVA), $n = 52$ tree islands) and dashed arrows indicate non-significant effects. Percentage values indicate explained variance of each endogenous variable. **b**–**d**, Effects on indicators of biodiversity quantified by species richness (**b**), Simpson diversity (**c**) and ecosystem functioning (**d**); direct effects of planted tree diversity are indicated by dark green bars and indirect effects through structural complexity are indicated by light green bars and the direct effects of tree island area are indicated by black bars and indirect effects through tree dominance are indicated by grey bars. The proportion of explained variance is shown to the left of **b** and **c**. All the bars indicate significant effects ($P < 0.05$; two-sided ANOVA, $n = 52$ tree islands). The legend for icons is presented in Fig. 2.

values within the study system[18] (here, within our landscape combining islands and conventionally managed oil palm monocultures). For example, at the 50% threshold, multidiversity increases by 1.5 in islands compared to conventionally managed oil palm monocultures. In other words, four and 2.5 biodiversity indicators reached at least 50% of their maximum observed species richness in tree islands and conventionally managed oil palm monocultures, respectively (Supplementary Table 4). Similarly, six and three ecosystem functioning indicators reached at least 50% of their maximum observed values in tree islands and conventionally managed oil palm monocultures, respectively (Supplementary Table 5). Overall, our results provide evidence of multidimensional ecological restoration benefits with tree islands in oil palm-dominated landscapes. Although the main priority is the protection of the remaining tropical forests[24], ecological restoration with tree islands along with other practices[7,25] and riparian buffer management[26,27] plays an essential and complementary role in safeguarding biodiversity and ecosystem functioning in cash crop-dominated landscapes.

Confirming our initial hypothesis, larger tree islands resulted in greater restoration benefits, both for ecosystem functioning (island area: $F = 12.9$, $P < 0.0001$; Extended Data Table 1) and biodiversity. Yet, the effects of island area on biodiversity varied across indicators (island area × indicator: $F = 5.1$, $P < 0.0001$ for species richness; $F = 2.8$, $P = 0.003$ for Shannon diversity; and $F = 1.8$, $P = 0.06$ for Simpson

diversity; Extended Data Fig. 1). Our structural equation models (SEMs) revealed that the influence of the island area acted through changes in tree dominance (Fig. 3a, Extended Data Table 2 and Supplementary Tables 6 and 7), with higher multidiversity in larger tree islands that are dominated by trees rather than oil palms. The higher tree dominance in the canopy and the thicker leaf litter (Extended Data Fig. 2 and Extended Data Table 3) might provide habitats of sufficient quality and quantity to enhance multidiversity. Large tree islands may thus act as keystone structures[28] in oil palm-dominated landscapes that facilitate the arrival of seed rain (especially of locally rarer species, see Fig. 3b) and the colonization, establishment and maintenance of diverse communities, such as understorey arthropods and trees (Fig. 3b,c). Although multifunctionality also increased with island area and the effect was mediated by changes in tree dominance, the strength of the effect depended on the method used to calculate multifunctionality (Extended Data Fig. 3, Extended Data Table 2 and Supplementary Table 7). When calculated for individual functions, large tree islands were pivotal for providing ecosystem functions related to predation and herbivory (through predatory arthropods and soil herbivores) and carbon and nutrient cycling (through decomposers; Fig 3d and Extended Data Table 3). By using constant sampling area or rarefaction curves, we could rule out that the influence of island area on biodiversity was limited to passive sampling[29] (Methods). Thus, ecological mechanisms associated with

environmental filtering such as reduced edge effects and greater environmental heterogeneity probably explain the positive effects of larger tree islands on multidimensional restoration benefits.

The effect of planted tree diversity on biodiversity—when considering abundances—depended on the biodiversity indicator (planted diversity × indicator: $F = 2.3$, $P = 0.014$ for Shannon diversity; $F = 2.8$, $P = 0.004$ for Simpson diversity; Extended Data Table 1). For example, planted tree diversity promoted the diversity of (non-planted) trees but decreased the diversity of arthropods (Extended Data Fig. 1). The statistically non-significant effects of multidiversity are probably due to contrasting responses of biodiversity indicators to planted tree diversity, with contrasting responses mediated by vegetation structure as shown by the SEM (Fig. 3b–d and Extended Data Table 2). Higher planted tree diversity led to structurally more complex habitats[30] (Extended Data Fig. 2) that benefited some biodiversity indicators (Shannon and Simpson diversity of bats and herbs), whereas others benefited from more open and structurally simpler habitats (for example, species richness of seeds and understorey arthropods; Fig. 3b,c and Supplementary Table 6). More open habitats also favoured native seeds, pollinators and predators (arthropods) and soil herbivores; whereas structural complexity enhanced above-ground biomass, litter input and microclimate buffering (Fig. 3d and Supplementary Table 7). Through changes in structural complexity, tree diversity had a negative impact on multifunctionality, although the strength of the effect depended on the methods of calculation (Extended Data Fig. 3, Extended Data Table 2 and Supplementary Table 7). Owing to specific responses associated with the adaptability of different organisms to contrasting habitats[31], establishing a combination of tree islands that differ in structural complexity may favour differences in local community composition leading to increases of gamma diversity and ecosystem functioning at the landscape scale[32].

Our study shows that the magnitude of the effect of ecological restoration on oil palm yield depends strongly on the spatial scale, with declines at the local scale, that is, within tree islands but no statistically significant reduction at the landscape scale. At the local scale, per area yields within tree islands were on average 24% lower than in the conventionally managed oil palm monocultures (Extended Data Fig. 4) because of the removal of oil palms, which reduced palm density (Extended Data Fig. 5b). In contrast, no statistically significant difference was detected in per island yield when including the yield of palms adjacent to tree islands (Fig. 2). The yield gains per oil palm surrounding the tree islands thus compensated for yield losses per area within the islands, with these beneficial effects resulting from oil palm thinning in the tree islands (Extended Data Fig. 5a,e). These beneficial effects were already observed a few years after establishment of the experiment[33] and are consistent regardless of the time period considered (Supplementary Note 1). Over time, yield decreases within tree islands are expected because of competition with trees, particularly in tree islands with higher planted diversity (Fig. 3d and Extended Data Fig. 4). Yet, these effects will remain negligible on industrial large-scale plantations because of the relatively small area covered by the tree islands. In our experiment, tree islands covered only 2.8 ha, less than 5% of the 140 ha industrial oil palm plantation. In contrast, smallholder oil palm plantations often only comprise a few hectares[5], of which (larger) tree islands would cover a more substantial proportion. In these cases, a decrease in yield may be compensated by extra goods from the tree islands, for example, fruits, natural latex, timber and firewood[34,35]. Furthermore, smallholders could benefit from higher levels of several ecosystem services, lower susceptibility to disturbance and risk diversification[17].

From an economic perspective, oil palm represents a highly profitable cash crop[6]. Consequently, replacing oil palms with native tree species typically raises concerns about high opportunity costs of lost revenue among landholders. Our large-scale study offers unique empirical evidence on the viability of multidimensional ecological benefits without compromising yield in oil palm-dominated landscapes by planting tree islands. To enhance the establishment of tree islands in oil palm- and other cash crop-dominated landscapes, they could be incorporated as a requirement in existing sustainability certifications (for example, Roundtable for Sustainable Palm Oil), alongside other practices including optimized management[25], ecological intensification and riparian restoration[26,36,37]. Enhancing the status of sustainability certifications should, however, not come at the expense of smallholder farmers, who are often excluded from certification programmes[38], nor at the expense of the protection of remaining intact forests for their exceptional value as refugia for biodiversity and providers of ecosystem functioning[39]. Overall, we provide robust evidence that biodiversity and ecosystem functioning in cash crop-dominated tropical landscapes can be enhanced without compromising overall agricultural productivity by planting tree islands. Although our study was conducted in a single landscape, it adds to growing experimental[37,40,41] and modelling evidence[42] of the ecological and economic benefits in oil palm agroforestry systems. Understanding how biodiversity and ecosystem functioning change in several landscapes[43,44] is urgently needed for designing and scaling-up ecological restoration of oil palm landscapes worldwide.

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

## Methods

### The biodiversity enrichment experiment

Our study was conducted in EFForTS-BEE, the biodiversity enrichment experiment of the EFForTS project (Ecological and Socioeconomic Functions of Tropical Lowland Rainforest Transformation Systems (Sumatra, Indonesia))[19]. EFForTS-BEE is part of the global network of tree diversity experiments TreeDivNet[45] (https://treedivnet.ugent.be/). The study region is characterized by a humid tropical climate with a mean temperature of $26.7 \pm 0.2\,°C$ and an annual rainfall of $2,235 \pm 381\,mm$ and the dominant soil type is loamy Acrisol[46]. In December 2013, 52 experimental plots (tree islands) were established in a conventionally managed 140 ha oil palm plantation. Following a random partition design[47], we systematically varied island area (25, 100, 400 and 1,600 $m^2$) and planted diversity (zero, one, two, three and six tree species). The six planted tree species (*Archidendron jiringa* (Jack) I.C.Nielsen (Fabaceae), *Parkia speciosa* Hassk (Fabaceae), *Durio zibethinus* L. (Malvaceae), *Dyera polyphylla* (Miq.) Steenis (Apocynaceae), *Shorea leprosula* Miq. (Dipterocarpaceae) and *Peronema canescens* Jack (Lamiaceae)) are native to the region and widely used for their fruits, timber or latex[35]. Around 40% of the oil palms located inside the tree islands were felled, with the number of felled oil palms differing depending on the tree island area[33]. The trees were planted between the felled and standing oil palms on a 2 m triangular grid. The tree islands were fenced and the management comprised a total stop of fertilizer, herbicide and pesticide application after planting. After May 2016, manual weeding was restricted to 1 m circles around the planted trees when these were shorter than the surrounding grass layer, allowing for natural regeneration. In addition to the 52 tree islands, we established four control plots in the oil palm plantation that have a fixed area (100 $m^2$) and that were managed conventionally (that is, no oil palm was felled, no tree was planted and application of fertilizer, herbicide and pesticide was as usual), in the main text referred to as conventionally oil palm monocultures. In total, the experiment comprises 56 study plots[19]. In each study plot larger than 25 $m^2$, one subplot of $5 \times 5\,m^2$ was established in a random location at a minimum distance of 1.5 m from the plot edge.

### Field measurements

We conducted an interdisciplinary field campaign from October 2016 to October 2018, that is, 33–57 months after establishment of the experiment. At this early stage of the experiment, the tree islands already differed in their structural complexity[30] and the planted trees reached up to 16 m height[35]. In all the 56 study plots, several indicators related to biodiversity, ecosystem functioning and structure were measured using standardized procedures and constant sampling areas at the level of the plot or subplot (Supplementary Tables 1–3). Only trees were sampled at unequal areas (that is, all trees present in the plots were sampled) and were therefore standardized using rarefaction curves (see 'Trees'). Oil palm yield was continuously measured since the beginning of the experiment at the level of individual palm but the data were then aggregated over space and time (see 'Per area yield' and 'Per island yield'). Each variable presented in the main text had one measurement per plot, such that blinding and randomization were not applicable. No statistical methods were used to predetermine sample size. The data were processed and analysed in R v.1.2.1335 (ref. 48).

### Birds and bats.

We recorded audible and ultrasound in March 2017 using automated sound recorders (SM2Bat+ recorders, Wildlife acoustics; with an acoustic SMX-II microphone on the left channel and one full-spectrum Sonitor Parus[49] microphone on the right channel), strapped to wooden poles at a height of 1.5 m in the centre of the plot. On consecutive days, we extracted sound recordings for sampling birds and insectivorous bats. We used two stereo 15 min recordings starting 15 min before sunset and two 15 min stereo recordings starting at sunrise, sampled at 22.05 kHz for birds. We used two 40 min mono sound recordings from the right channel, extracted from consecutive nights, starting 20 min after sunset, sampled at 384 kHz for bats. Twelve sound recorders were installed simultaneously in 12 randomly chosen plots. The recordings were annotated in ecoSound-web[50] to extract the duration of each bird vocalization and bat pass and bird detection distances were estimated using reference sound transmission sequences[51]. We assigned birds to species according to Birdlife International taxonomy. Owing to the lack of standard protocols and reference collections for Southeast Asia, we could not identify bats to species and used sonotypes instead. We appended feeding guild information to each bird species[52] all detectable bats were echolocating and thus considered insectivorous bats. Only bird vocalizations detected within a 28 m radius were included, which corresponds to the diameter of the largest study plot ($40 \times 40\,m^2$). We used the maximum number of individuals detected simultaneously in all recordings per plot as a conservative proxy of abundance per bird species or bat sonotype.

### Understorey arthropods.

Arthropods were sampled in the understorey vegetation during October 2016 to January 2017. Each plot was sampled three times with six pan traps per plot exposed for 45 h. Traps were made of white plastic soup bowls covered with yellow ultraviolet spray-paint[53] and were filled with water and one drop of regular soap. They were fixed in a holding system in groups of three at the height of the surrounding plants and these systems then equally distributed in distance from edge and to each other. All arthropods were preserved in 70% ethanol. Subsequently, all individuals were identified to higher taxonomic groups and morphospecies. The taxonomic groups Hymenoptera, Lepidoptera and Araneae were categorized into functional groups (pollinators, predators and parasitoids) using different identification keys[54–60] (Supplementary Table 10). Predators and parasitoids were merged into the single functional group 'predators'.

### Soil fauna.

During October–November 2016, in each plot, four soil samples of $16 \times 16\,cm^2$ were taken randomly within the subplot with a spade. Samples included litter (if present) and soil down to a depth of 5 cm. Animals were extracted using a gradient heat extractor[61] and collected in dimethyleneglycol–water solution (1:1) and thereafter transferred into 70% ethanol[62]. All extracted animals were counted and sorted into 28 taxonomic groups (in most cases orders) allowing for functional group classifications[63]; Extended Data Table 4. We calculated community metabolism of all animals that were classified as detritivores, herbivores and predators in a sample by using mean group- and ecosystem-specific estimates derived from ref. 63. The estimates are based on measurements of more than 5,000 individuals of soil animals across eight different oil palm plantations in the same region; to estimate community metabolism, individual body masses were recalculated to metabolic rates using group-specific regressions from ref. 64. Community metabolism was calculated by summing up metabolic rates of all individuals; we used the mean per plot across four samples for each functional group (detritivores, herbivores and predators) for the analysis. We also computed taxonomic diversity as the number of taxonomic groups (in most cases orders) present in each plot for the analysis.

### Fungi.

In January 2017 three soil cores (10 cm depth, 4 cm diameter) were taken within each $5 \times 5\,m^2$ subplot. Surface leaf litter was removed before soil collection. The soil was sieved through a $50 \times 50\,mm^2$ sieve and roots were separated from soil. The fungal community was assessed using Illumina next-generation sequencing (Illumina) of the ITS2 marker region. The detailed protocol for amplification, amplicon sequencing and generation of fungal operational taxonomic units (OTU) is described in ref. 65. OTUs were classified taxonomically using the BLAST (blastn, v.2.7.1) algorithm[66] and the UNITE v.7.2 (UNITE_public_01.12.2017.fasta) reference database[67].

**Prokaryotes.** In May 2017 three cores of topsoil (10 cm depth) were taken in each subplot. Soil cores were then mixed, homogenized and freed from roots. A total of 5 ml of RNAprotect Bacteria reagent (Qiagen) was added to 5 g of soils to prevent nucleic acid degradation. DNA and RNA were extracted from 1 g of soil by using the Qiagen RNeasy Power-Soil Total RNA Kit and the RNeasy PowerSoil DNA Elution Kit (Qiagen). The V3–V4 region of the 16S ribosomal RNA gene was amplified and sequenced as described in ref. 68. Paired-end sequences were quality filtered with fastp (v.0.20.0)[69] and merged with PEAR v.0.9.11 (ref. 70). Remaining primer sequences were clipped with cutadapt v.2.5 (ref. 71). Size filtering, dereplication, denoising and chimaera removal was performed with vsearch v.2.12.0 (ref. 72). Curated sequences were then classified by mapping each sequence against the SILVA database with the BLAST[73]. Counts were normalized by using the GMPR normalization[74].

**Seeds.** We installed four seed traps in each of the 56 study plots for 1 yr, that is, between 1 April 2017 and 29 March 2018. The traps were built using fine-mesh cloth attached to a squared structure made in PVC pipes of size 50 cm × 50 cm fixed at 1 m from the ground. The traps were installed at random locations in each of the four quadrants of each plot, at a minimum distance of 1 m from the plot edge. The contents of the traps were collected twice a month, dried at approximately 40 °C during 3–7 days. All the seeds were carefully extracted from the samples, counted and separated by morphospecies using hand lenses (×10 magnification) and a microscope (Leica photomicroscope with ×400 magnification) for very small seeds. Molecular identification of the morphospecies was implemented using three universal plant DNA barcodes (*matK, rbcL* and *ITS2*)[75–78] and taxonomic assignments were made using BLASTn search against the NCBI Genbank reference sequence database[79]. Sequences obtained from the barcode loci were deposited in NCBI Genbank under the accession numbers OM811991–OM812021, OM837673–OM837724 and OM935782–OM935815. We classified each morphotype as native or non-native species using available literature[80,81] (http://www.plantsoftheworldonline.org/). We derived the native seed density (number of native seeds per m$^2$) as the total number of native seeds over the entire sampling duration per plot, which was used as an indicator of ecosystem functioning (see 'Ecosystem functioning'). Seed diversity, calculated on the basis of the Hill number frameworks and used as indicators of biodiversity (see 'Biodiversity'), was derived from the pooled samples per plot over the entire sampling duration for all seeds (native and non-native).

**Herbs.** All non-woody terrestrial vascular plants (for example, angiosperm herbs and vines, ferns, but not epiphytes) in the subplot were recorded from February until March 2018. They were classified to species or morphospecies and herbaceous cover (in absolute per cent ratios from 1% to 100%) was estimated by two people. Epiphytes growing on the stems of trees or palms were excluded, whereas vine species that rooted in the ground and climbed up stems of trees or palms were included. Herbarium specimens were collected and stored in the laboratory of Jambi University. All names were checked following The Plant List 2013, v.1.1 (http://www.theplantlist.org).

**Trees.** All planted trees were surveyed in January to February 2018 as part of a yearly inventory[35]. Furthermore, we surveyed all free-standing woody plants (trees, shrubs and bamboos) that colonized the plots with a length of ≥130 cm from April until August 2018. For each species or morphospecies, one voucher specimen was collected, dried and pressed according to standard procedure. In the main text, we refer to the colonized woody plants as 'trees', unless stated otherwise. Because the number of sampled trees largely varied according to the tree island area, we standardized the diversity estimates using rarefaction curves (R package iNEXT)[82] to 24 individuals, which represent the median number of individuals per plot.

**Pollen.** To collect pollen/spore rain, Behling pollen traps[83] were installed from June until October 2018. Each trap consists of a plastic tube which is placed about 30 cm above the ground and is held by a fixing pole. The tube is filled with 5 ml of liquid glycerol, synthetic cotton and, on the top, it is covered by a mosquito net to reduce disturbance from animals or litter and prevent the cotton from being removed. In tropical regions heavy rainfalls occur, thus it is necessary to prevent the pollen from pouring out of the pollen trap. In the Behling trap, glycerol is used, which has a higher density compared to water. Consequently, the incoming rainfall can flow out of the trap without taking the pollen, which is trapped in the synthetic cotton and in the glycerol[83]. The Behling traps were modified to mimic the surrounding environment and maximize recovery. In total, 168 pollen traps were installed in the plots (3× plot). Of the total 56 plots, the pollen traps were not recovered in three plots (P28, P34 and P47). One pollen trap from each 53 plot was processed and analysed. Firstly, each pollen trap was washed with distilled water through a 2 mm mesh sieve to remove large size materials. Afterwards, the pollen traps were sieved through a 150 μm mesh sieve to exclude medium-sized materials from the samples. Two *Lycopodium* tablets were added as markers to each sample to estimate palynomorph concentrations[84] and the Erdtman acetolysis[85] was applied, to remove cellulose material. Residues were mounted in glycerol jelly for pollen visualization, identification and counting. Pollen and spore analyses were carried out using light microscopy. All identified pollen and spore types were photographed using a Leica photomicroscope with ×400 magnification. For each trap, a total sum of at least 100 pollen grains were counted. Pollen and spore grains can rarely be identified to species level and the level of taxonomic identification varies for different groups of plants. Consequently, a reduction to the family level has been proposed for studies involving analysis of palynological diversity in the tropics[86].

**Pollination.** We assessed pollination rate on chilli pepper plants (*Capsicum annuum*) as phytometer plants, selected for potential shade tolerance[87], widespread home garden cultivation in this region[88] and the potential role pollination can play in fruit quality and yield[89]. We raised 1,500 individuals of a locally available variety of *C. annuum* from seed outside of the study plots. During the growth period outside the study plots, we applied NPK fertilizer and pesticide (imidacloprid, deltamethrin, mancozeb and abamectin) following local practices to standardize growing conditions and control pest damage before transfer to field sites. We halted fertilizer and pesticide application 1 week before placement in the plots and only watered as conditions required thereafter. In February 2018, we selected 224 healthy individuals of comparable size to transfer to the 56 study plots (four plants per plot). The four chilli plants were placed, still in their pots, at the centre of each plot for 5 weeks for a period of open pollination and monitoring, followed by 3 weeks for fruit harvesting. We removed any flowers before placement in the field, so pollinated flowers and developing fruits were assumed to result from pollination within the study plots. During the period before final harvest, each plot was revisited once per week and the number of flowers were counted per plant. The rate of successful pollination was estimated from the fruit to flower ratio, which was the total number of harvested fruits divided by the total number of flowers observed per plot.

**Per palm yield.** We followed the conventional harvesting procedure established by the plantation manager of PT Humusindo and measured the weight of the fresh fruit bunches directly after harvest using a portable scale. We measured the per palm yield (kg per palm) of all palms inside the 52 tree islands ($N$ = 214) and one palm per control plot with conventional management ($N$ = 4). To obtain a more solid estimate for the conventional plantation, we measured the per palm yield of 30 more reference palms that were evenly distributed across the conventional

plantation at approximately an equal distance to each tree island and whose neighbourhood is characteristic of conventionally managed oil palm monocultures (Supplementary Note 2 and Supplementary Figs. 1–3). To examine potential changes in yield in the conventionally managed oil palm plantation surrounding the tree islands ('spillover effects'), we measured the per palm yield of three oil palms adjacent to each tree island, at increasing distance to the island's edge (at position number 1, 2 and 3)[33]. For direct comparison with earlier findings, we analysed the data following established methodology[33]. The tests were based on linear mixed-effect models with the annual yield of individual oil palms as the response variable and the plot identity as the random effect. Pairwise comparison was conducted with a post hoc Tukey test. Because our results indicate that only the palm directly adjacent to the tree island (in position number 1) was affected by the experimental treatment (Extended Data Fig. 5a, in agreement with ref. 33), we do not consider the palms in position 2 and 3 in the yield calculation per island (Per island yield). The per palm yields in and adjacent to the 56 study plots have been continuously monitored since the establishment of the experiment in December 2013. The extra 30 palms were established in December 2016. For consistency with other indicators (Extended Data Tables 1–3), we reported yield data for 1 yr (November 2017 until October 2018) in the main text and for 2 yr (November 2016 until October 2018) in the Supplementary Note 1. Yield data since 2014 are shown in Extended Data Fig. 4, in which the oil palms in position 3 were used as reference palms for the corresponding time period.

**Per area yield.** We estimated per area yield ($\Delta Y_{ha}$, kg ha$^{-1}$) as the yield of a given palm (kg palm$^{-1}$) multiplied by a stand density-dependent expansion factor (EF) to derive estimates of per area yield (kg ha$^{-1}$). We then calculated the per area yield change between tree islands and reference (kg ha$^{-1}$; Supplementary Note 3). This approach accounts for changes in per area yield due to oil palm thinning (that is, reduced oil palm densities and changes in per palm yield in the tree islands) but does not account for potential changes in per palm yield on the surrounding plantation, for example, because of spillover effects[19]. An alternative analysis considering spillover effects was performed at the plot level (Per island yield).

**Per island yield.** We estimated oil palm yield changes at the tree island scale ($\Delta Y_{Island}$, in kg island$^{-1}$); equations (1)–(4) following established methodology[33]. This method considers the yield foregone owing to the removal of some oil palms before the experiment, as well as changes in per palm yield inside the tree islands and directly adjacent to the tree islands (at position 1, that is, spillover effects). Because the number of oil palms inside and adjacent to the tree islands and the number of removed oil palms vary depending on tree island area[33], the net oil palm yield changes are provided per plot and not per area. Even though this method was initially designed to calculate the oil palm yield changes for the tree islands, here we also apply it to the four control plots to integrate them in our synthesis analysis.

$$\Delta Y_{Island} = Y_{Spillover} + Y_{RemainChange} - Y_{Foregone} \qquad (1)$$

$$Y_{Foregone} = N_{felled} \times Y_{p\_ref} \qquad (2)$$

$$Y_{RemainChange} = N_{in} \times (Y_{p\_in} - Y_{p\_ref}) \qquad (3)$$

$$Y_{Spillover} = N_{adj} \times (Y_{p\_adj} - Y_{p\_ref}) \qquad (4)$$

$\Delta Y_{Island}$, per island oil palm yield change (kg island$^{-1}$)
$Y_{Spillover}$, per island yield changes due to spillover effects (kg island$^{-1}$)
$Y_{RemainChange}$, per island yield changes inside the island (kg island$^{-1}$)
$Y_{Foregone}$, per island yield foregone due to oil palm removal (kg island$^{-1}$)
$N_i$, number of remaining oil palms inside the island

$N_{ad}$, number of oil palms directly adjacent to the island (that is, adjacent position 1)
$N_{felled}$, number of removed oil palms in the island
$Y_{p\_ref}$, median per palm yield of the reference palms in the conventionally managed oil palm plantation (kg palm$^{-1}$)
$Y_{p\_in}$ median per palm yield inside the tree island (kg palm$^{-1}$)
$Y_{p\_adj}$, per palm yield directly adjacent to the tree island (that is, adjacent position 1 (kg palm$^{-1}$)).

**Above-ground biomass.** For all the planted trees, we measured the basal diameter (at 10 cm above ground), the diameter at breast height (130 cm above ground) and the tree height in 2017 as part of a yearly inventory[35]. In January and February 2017, we also measured the height of the oil palms at meristem, that is, the point of attachment of the young leaves to the oil palm trunk[30]. We estimated above-ground biomass of the trees (equation (5)) and the oil palms (equation (6)) using the respective allometric equations of refs. 90,91:

$$AGB_{tree} = 0.0673 \times (\rho \times DBH^2 \times H)^{0.976} \qquad (5)$$

$$AGB_{palm} = 71.797 \times H - 7.0872 \qquad (6)$$

$AGB_{tree}$, above-ground biomass of the planted trees (kg tree$^{-1}$)
$AGB_{palm}$, above-ground biomass of the oil palms (kg palm$^{-1}$)
DBH, tree diameter at breast height (cm)
$H$, height of tree or palm (m)
$\rho$, wood density (g cm$^{-3}$).

Wood density for *Peronema canescens* (0.61 g cm$^{-3}$), *Parkia speciosa* (0.54 g cm$^{-3}$) and *Dyera polyphylla* (0.36 g cm$^{-3}$) was based on EFForTS core plot data, whereas for *Archidendron* sp. (0.36 g cm$^{-3}$), *Shorea leprosula* (0.44 g cm$^{-3}$) and *Durio zibethinus* (0.516 g cm$^{-3}$) it was taken from the global wood density database[92].

We estimated the total above-ground biomass per plot as the sum of the above-ground biomass of the palms and the planted trees (equation (7)). The estimations of total AGB did not consider the necromass, litter, understorey vegetation and spontaneously established trees, which were considered negligible.

$$AGB = (\sum AGB_{tree} + \sum AGB_{palm}/N_{in} \times d_{palm})/(A \times 10,000) \qquad (7)$$

AGB, total above-ground biomass per plot (t ha$^{-1}$)
$d_{palm}$, density of oil palms (number of oil palms per ha) that takes into account the local neighbourhoods of the plots (also referred to as EF; see Supplementary Note 3)
$A$, area of the plot (m$^2$).

**Tree growth.** The growth of the planted trees per plot was calculated as[35]:

$$BA_{inc,2017-2018} = \sum (BA_{tree,2018} - \sum BA_{tree,2017})/A. \qquad (8)$$

$BA_{inc,2017-2018}$, total plot-level basal area increment between 2017 and 2018 (in cm$^2$ m$^{-2}$ yr$^{-1}$, equivalent to m$^2$ ha$^{-1}$ yr$^{-1}$)
$BA_{tree,year}$, tree basal area (in cm$^2$) derived from the basal diameter (cm) in the specific year.

**Leaf litter input.** We measured leaf litter fall (in g m$^{-2}$ yr$^{-1}$) using the four seed traps installed randomly in each four quadrants of the plots from April 2017 to March 2018 (Seeds). The contents of the traps were collected twice a month, dried at 40 °C for 4–7 days and weighted. We also sorted the leaves by species and weighted the content for the six planted tree species and oil palm separately.

For each sampling date, we aggregated the values at plot level using the median per plot of the litter weight. We then excluded outliers defined as plot-level values outside the range of 3 standard deviations

around the median of the entire data (less than 5% of the litter weight data, total and per species). To get annual estimates, we summed the available plot-level values over time and divided them by the number of sampling dates (that is, between 17 and 24, depending on the number of missing traps or excluded outliers). We then multiplied the obtained values by the seed trap area (0.25 m$^2$) to get the leaf litter fall in g m$^2$ yr.

**Leaf litter decomposition.** We installed litterbags (20 × 20 cm$^2$, 4 mm mesh size) each filled with 12 g of material: 6 g of freshly cut and air-dried (approximately 25 °C) fronds of oil palm leaves[93] and 6 g of the freshly fallen air-dried leaf litter for each tree species or their combinations in experimental plot. In each plot, one litterbag was installed in November 2017 for a duration of 6 months. Decomposition (litter mass loss) was calculated as the difference between the initial litter dry mass and litter dry mass remaining after 6 months and expressed as a percentage of decomposed material.

**Water infiltration capacity.** To quantify soil water infiltration capacity, we measured saturated soil hydraulic conductivity ($K_{fs}$, cm h$^{-1}$) using a dual-head infiltrometer (Saturo) in March 2018 near the subplot centre in 35 (out of the 52) tree islands and in the four control plots representing conventionally managed oil palm monocultures. Owing to a broken instrument, the 17 remaining plots were measured using a custom manual double-ring infiltrometer, which tends to yield higher $K_{fs}$ estimates than the dual-head approach because there is no correction for lateral flow. In three plots, $K_{fs}$ was measured with both devices. We plotted these values against each other and found a close linear relationship ($R^2 = 0.98$, $P = 0.066$); even though it was only marginally significant because of the small sample size, we used it to correct the values from the 17 plots that were measured manually ($K_{fs\_corr} = 1.44 + 0.55\, K_{fs\_double\_ring}$) to allow for comparability across all 56 plots.

**Evapotranspiration.** We recorded land and canopy surface temperatures using a radiometric thermal camera (FLIR Tau 2 640, FLIR Systems) attached to a TeAx ThermalCapture module (TeAx Technology GmbH) mounted on a multicopter drone (MK EASY Okto V3; HiSystemsy) as described in ref. 94. Image sets were recorded four to five times per day around noon, covering each plot once over a 9 day period encompassing varying weather conditions. Land and canopy surface temperatures were the main input for modelling latent heat flux (in W m$^{-2}$) and deriving evapotranspiration using the QWaterModel QGIS3 Plugin[95], which is based on the DATTUTDUT energy balance model[96]. Measured short-wave radiation and relative humidity were used as further input variables to support the prediction of latent heat flux and derive evapotranspiration.

**Microclimate.** We measured microclimate using temperature per humidity loggers: hydrochron (DS1923-F5) and thermochron (DS1922L-F5) iButtons, Maxim integrated. The loggers were installed in the middle of each plot at 1.5 m above ground and were protected from water and direct solar radiation using handmade multiplate radiation shields[97]. Data were collected for 1 yr (18 November 2017 until 19 September 2018) every 3 h starting at midnight. As a proxy for microclimate buffering, we calculated the daily amplitude as the absolute difference between values at 7:00 and 15:00 (ref. 97), aggregated using the median value over the entire measurement period.

**Soil properties.** We determined soil total carbon content (g mg$^{-1}$), total nitrogen content (mg g$^{-1}$) and plant available phosphorus content (mg g$^{-1}$) using the same three soil samples as for fungi community data (see 'Fungi') and the method of determination is described in detail in ref. 65. We then calculated the C:N ratio accounting for the molar mass of the elements following ref. 98, that is, 12.0107 for carbon and 14.0067 for nitrogen. We also measured soil bulk density (g cm$^{-3}$) using five soil samples taken in the subplot in May 2018. Soil rings of 100 cm$^3$

were inserted horizontally into the first 5 cm of topsoil. The soil was weighed, dried at 105 °C until constant weight and weighted again. Calculation was done on the dry weight basis, for which the sample dry weight (g) was divided by the volume of the sample (cm$^3$) collected from the average of the five replicates. We used the mean per plot for all mentioned soil variables and used the inverse of C:N ratio and soil bulk density as measures of soil fertility and soil decompaction, respectively (Supplementary Table 1).

**Vegetation structure.** We measured 12 variables representing various aspects of the vegetation structure (Supplementary Table 3). We used a terrestrial laser scanner Focus M70 (Faro Technologies) to create three-dimensional point clouds of the vegetation at the centre of each plot in September and October 2016, as described in ref. 30. We computed the (1) stand structural complexity index (SSCI) following ref. 99 and its two components: (2) the mean fractal dimension index (MeanFRAC) derived from cross-sections of polygons in the three-dimensional point cloud, which is a scale-independent and density-dependent measure of structural complexity and (3) the effective number of layers (ENL) that describes vertical stratification based on the Simpson Index[100]. ENL and MeanFRAC are integrated in the SSCI and all these three measures were derived from vegetation parts above 130 cm. We also derived (4) the understory complexity index that measures the fractal dimension of horizontal cross-sections of the point cloud between 80 and 180 cm height, thereby measuring the structural complexity of the understorey vegetation[101]. (5) Canopy gap fraction was estimated from hemispherical photographs at plot level as described in ref. 30. Drone-based photogrammetry dated from September to October 2016 was used to further partition the canopy (in %) as (6) oil palm cover and (7) tree cover as described in ref. 102. We also used the drone-based orthophotos to calculate (8) oil palm density as the number of living oil palms per plot irrespective of the orientation of the plot relative to the planting scheme (Supplementary Figs. 2 and 3) . For the smaller plots (5 × 5 m) unaffected by thinning, the oil palm density was simply the typical planting density in conventionally managed oil palm plantations (120 planted palms per hectare). Further details on the oil palm density calculation are given in the Supplementary Note 3. We also calculated (9) tree density as the number of trees planted and from natural regeneration per plot and expressed per hectare. We estimated the portion of the ground (in percent) as (10) understorey vegetation cover and (11) litter cover per subplot in February–March 2018. The understorey vegetation cover included all parts of plants lower than 130 cm in height, including the trunks and other parts of the planted trees but excluding oil palm trunks. (12) The litter depth (cm) was measured as the mean value in three randomly chosen positions inside each subplot with a metal ruler. To extract orthogonal axes (PC1 and PC2) that represent most of the variability in the vegetation structure, we applied a principal component analysis on all the structural variables after standardization to zero mean and unit variance.

### Restoration outcomes

**Ecosystem functioning.** We measured 20 variables related to seven categories of ecosystem functioning including: productivity as (0) tree growth (basal area increment of the planted trees in m$^2$ ha$^{-1}$ yr$^{-1}$) that was further excluded from the analysis—see Supplementary Fig. 4, (1) oil palm yield (per island oil palm yield changes in kg of fresh fruit bunches per island) and (2) above-ground biomass (biomass stored in the aerial parts of the planted trees and the oil palms, in t ha$^{-1}$); resistance to invasion as (3) native seeds (total number of arriving native seeds per m$^2$) and (4) resistance to invasive plants (100−observed cover of *Clidemia hirta*, in %); pollination as (5) pollinators (number of sampled individuals) and (6) pollination rate (fraction of flowers on phytometer plants that are pollinated, %); soil quality as (7) soil P (phosphorous content, %), (8) 1/soil C:N (the molar ratio of soil C to soil N concentration) and (9) soil decompaction (inverse of soil bulk density in g cm$^{-3}$); predation

and herbivory as (10) predatory invertebrates (total activity duration of insectivorous bats and birds, in seconds); (11) predatory arthropods (number of sampled individuals), (12) predatory soil fauna (energy flux, in J h$^{-1}$), (13) herbivory (energy flux, in J h$^{-1}$); carbon and nutrient cycling as (14) decomposers (energy flux, in J h$^{-1}$); (15) litter decomposition (relative biomass loss of litter after 6 months in litterbags, %) and (16) litter input (biomass of leaf litter falling in traps, g m$^{-2}$); water and climate regulation as (17) evapotranspiration (canopy latent heat flux, in W m$^{-2}$); (18) soil water infiltration capacity (saturated soil hydraulic conductivity in cm h$^{-1}$) and (19) microclimate buffering (median daily amplitude of air temperature during 1 yr, °C d$^{-1}$). A more detailed summary of the 20 ecosystem functioning variables is presented in Supplementary Table 2.

**Biodiversity.** We derived taxonomic diversity for soil bacteria and soil fungi, soil fauna, herbs, trees, seeds, pollen, understorey arthropods, birds and bats. Most of the groups (arthropods, herbs, trees, birds and seeds) were sorted at the lowest possible taxonomic level (species or morphospecies). Pollen, soil fauna and bats were sorted to higher levels, mainly family, order and morphotypes, respectively. Soil bacteria and soil fungi were analysed by DNA-based marker gene sequencing as amplicons sequence variants or OTU, respectively. Hereafter, we refer to these different taxonomic units (species, family, order, morphotypes and OTU) as 'species' for simplicity.

Diversity was measured following the Hill number framework, which allows comparison across diversity indices that weigh relative abundances to varying extents (species richness, Shannon diversity and Simpson diversity) and are expressed in terms of effective numbers of species[103–106]. Species richness is more sensitive to locally rare species, Simpson diversity is more sensitive to locally dominant species and Shannon diversity favours neither rare nor dominant species. We show results for species richness and Simpson diversity in the main text and for all indicators in the Extended Data Tables 1 and 2 and Supplementary Tables 4–8. The calculations were performed using the R packages iNext[82] and vegan[107].

**Multidiversity and multifunctionality.** Different indicators of biodiversity and ecosystem functioning were aggregated by calculating multidiversity and multifunctionality, respectively. Following ref. 18, we performed a cluster analysis to preselect indicators for achieving a representative measure of 'ecosystem function multifunctionality'. As tree growth and litter input were correlated and formed a cluster, we excluded tree growth from the analysis (Supplementary Note 4 and Supplementary Fig. 4). Following a threshold approach[108], we calculated multifunctionality (and multidiversity) as the number of ecosystem functioning (and biodiversity) indicators that cross a threshold, expressed as a certain percentage of the maximum observed values in our study landscape (among all 56 study plots). We calculated multifunctionality and multidiversity for all thresholds from 1% to 99% and presented results for a 50% threshold in the main text. To reduce the influence of extreme values, we used the mean of the three highest values observed in all study plots, respectively. As an alternative to the threshold approach, we also calculated multidiversity and multifunctionality as the average of the indicators[108]. Before multidiversity and multifunctionality calculations, all the variables were standardized to unit scale (for biodiversity and ecosystem functioning separately). The calculations were performed using the package multifunc in R[108].

## Statistical analysis

**Linear mixed-effect models.** We used linear mixed-effect models to test the effects of the experimental treatment on restoration outcomes. We fitted three separated models for biodiversity using species richness, Shannon and Simpson diversity as response variables and one model for ecosystem functioning. These models included tree island (compared to our controls of conventionally managed oil palm monocultures), island area (plot edge length in m), planted diversity and the restoration outcome (either biodiversity or ecosystem functioning indicators) as single factors and tree island × indicator, island area × indicator, planted diversity × indicator, island area × planted diversity and island area × planted diversity × indicator interactions. For conventionally managed oil palm plots, island area was set to 10 m edge length and planted diversity to zero. Each response variable (biodiversity and ecosystem functioning indicators) was standardized to unit scale (between 0 and 1) as this improved the model diagnostics before applying the respective linear mixed-effect models; whereas we used logarithmic transformations for island area and planted diversity. Plot was included as a random term.

As an alternative to the linear mixed-effect models, we applied Kruskal–Wallis tests on each indicator of biodiversity and ecosystem functioning for comparison between the 52 tree islands and the four conventionally managed oil palm monocultures as control plots (Supplementary Note 5 and Supplementary Tables 8 and 9).

**Structural equation modelling.** We used piecewiseSEM[109] to assess the influence of tree island area and tree planted diversity on biodiversity and ecosystem functioning operating through increasing tree dominance, through differences in structural complexity (indirect effects) or through alternative mechanisms (direct effects). As a hypothetical causal model, we included direct paths between island area and tree dominance (PC2) and between tree planted diversity and structural complexity gradient (from open to dense and structurally complex vegetation, PC1; Extended Data Fig. 2). Piecewise SEMs are based on a set of linear equations which are evaluated individually[109]. For our analyses, we included:

lm(restoration outcome ≈ island area + planted diversity) (1)
lm(structural complexity ≈ planted diversity) (2)
lm(tree dominance ≈ island area) (3)

Across all restoration outcomes, the main variables were always included in the linear model (1). As tree dominance and structural complexity are potential mechanistic pathways explaining the influence of island area and tree planted diversity, alternative paths between them and biodiversity or ecosystem functioning were added, if they improved the model fit (based on modification indices, $P < 0.05$). Therefore, model selection influenced only the inclusion of structural complexity and tree dominance in the linear model (1). Effects of island area and planted diversity through mechanistic pathways were calculated by multiplying their effect on the mechanistic explanatory variable and the effect of the mechanistic explanatory variable on biodiversity or ecosystem functioning. Mechanisms that were not captured by either of our proposed mechanistic pathways are represented by the direct paths between island area and tree planted diversity and biodiversity or ecosystem functioning. We tested the assumption of normality of the residuals in models (1), (2) and (3) using Shapiro–Wilk normality test, applied a suitable transformation of the response variables if needed (package bestNormalize v.1.6.1). The transformation concerned four out of ten indicators for species richness and Shannon diversity, three indicators out of ten indicators for Simpson diversity and 14 indicators out of 19 for ecosystem functioning. Effect sizes were calculated using standardized coefficients. The island area and planted tree diversity were log-transformed as this improved the model fit. For each SEM, we quantified the goodness of fit using the following metrics: Fisher's $C$ statistic and significance value based on a Chi-square test, the information criterion (Akaike information criterion (AIC), Bayesian information criterion (BIC), corrected AIC (AICc)) and pseudo-$R^2$ values and applied the test of directed separation as implemented in the package piecewiseSEM v.2.1.0[109].

## Inclusion and ethics

The research included researchers from the Indonesian institutes Jambi University and Bogor Agricultural University throughout the research

process—study design, study implementation, data ownership, intellectual property and authorship of publications. Local and regional research relevant to our study was considered in citations.

## Reporting summary

Further information on research design is available in the Nature Portfolio Reporting Summary linked to this article.

## Data availability

The raw data are available at https://data.goettingen-research-online. de/dataverse/crc990, with the specific link for each dataset provided in Supplementary Tables 1–3. The processed data are available at https:// doi.org/10.6084/m9.figshare.22320490. Seed DNA sequences are available in NCBI Genbank under the accession numbers OM811991–OM812021, OM837673–OM837724 and OM935782–OM935815. Sequencing data of the soil fungal community were deposited in the NCBI Sequence Read Archive (SRA) under Bioproject accession number PRJNA659225. The public UNITE database (https://unite.ut.ee/) v.7.2 on fungal ITS sequences was used as a reference of taxonomic classification. Sequence data of the bacterial communities were deposited in the NCBI SRA under Bioproject accession number PRJNA841353. Sequence identification was performed by mapping all curated sequences against the SILVA database v.132 (https://www.arb-silva.de/).

## Code availability

The R code used in the current study is available at https://doi.org/ 10.6084/m9.figshare.22320490.

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

**Acknowledgements** We thank PT Humusindo for granting us access to and use of their properties, as well as A. F. C. Rodríguez, A. Angulo, L. S. Donfack, M. Ehbrecht and D. Seidel for their contribution to data; the field assistants E. J. Siahaan, K. H. Dalimunthe and Juliandi for plot maintenance and for their technical assistance in data collection; and M. Teuscher and A. Gérard for their contribution to establishing the experiment. Illustrations were provided by F. Arndt, Formenorm Leipzig. This study was conducted using the research permit nos. 337/SIP/FRP/E5/Dit.KI/IX/2016 (D.C.Z.), 58/SIP/IV/FR/10/2021 and 20/SIP/IV/FR/1/2022 (C.M.), 53/EXT/SIP/FRP/E5/Dit.KI/IX/2017 (V.K.), 2C13UA0011-S (H.L.), 54/EXT/SIP/FRP/E5/Dit.KI/IX/2017 (A. Potapov), 4233/FRP/E5/Dit.KI/XII/2017 (L. Sachsenmaier), 343/SIP/FRP/E5/Dit.KI/X/2016 (A.K.), SK.336/KSDAE/SET/KSA.2/7/2019 (S.B. and C.A.S.), 1837/FRP/E5/Dit.KI/VIII/2017 (K.K.) and 1092/FRP/SM/VIII/2015 and 1487/FRP/SM/KI VI/2016 (W.K.) from the Indonesia Ministry of Research and Technology. The experiment used in this study is part of the global network of tree diversity experiments TreeDivNet (https://treedivnet.ugent.be/). Deutsche Forschungsgemeinschaft (DFG, German Research Foundation)—project ID 192626868—SFB 990 and the Ministry of Research, Technology and Higher Education (Ristekdikti) in the framework of the collaborative German–Indonesian research project CRC990 (D.C.Z., F.B., K.D., I.G., A. Potapov, A.R., I.A., J.B., H.B., D. Berkelmann, S.B., D. Buchori, R.D., O.G., F.E., R.F., N.H., B.I., W.K., V.K., A.K., J.K., K.L., H.L., M.M., M.S.M., C.C.M.M., Y.A.M., G.B.P., H.D.P., A. Polle, D.A.P., L. Sachsenmaier, S.S., F.S., A.C.S., L. Sundawati, T.T., M.W., D.H. and H.K.). Dorothea Schlözer Postdoctoral Programme of the Georg-August-Universität Göttingen (N.G.-R.).

**Author contributions** D.C.Z., N.G.-R., H.K. and D.H. conceived this work. D.C.Z. and N.G.-R. developed the methodology. D.C.Z., N.G.-R. and M.S.M. created the software. D.C.Z., N.G.-R. and H.L. undertook the formal analysis. D.C.Z., F.B., K.D., A. Potapov, I.A., J.B., D. Berkelmann, S.B., F.E., R.F., N.H., W.K., V.K., A.K., J.K., K.L., H.L., C.C.M.M., H.D.P., D.A.P., L. Sachsenmaier, F.S. and C.A.S. conducted the investigations. D.S. obtained resources. D.C.Z. undertook data curation. D.C.Z., N.G.-R. and H.L. were responsible for visualization. H.K., D.H., M.W., I.G., H.B., D. Buchori, R.D., O.G., M.M., Y.A.M., A. Polle, S.S., T.T., L. Sundawati and B.I. obtained funding. D.C.Z., L. Sundawati and B.I. conducted the project administration. H.K., D.H., I.G., H.B., D. Buchori, R.D., O.G., M.M., Y.A.M., A. Polle, S.S., T.T., L. Sundawati, B.I. and A.R. were responsible for supervision. D.C.Z. and N.G.-R. wrote the original draft. F.B., K.D., I.G., A.P., A.R., G.B.P., D.C., M.W., H.K. and D.H. reviewed and edited the final manuscript.

**Funding** Open access funding provided by University of Neuchâtel.

**Competing interests** The authors declare no competing interests.

**Additional information**
**Correspondence and requests for materials** should be addressed to Delphine Clara Zemp.

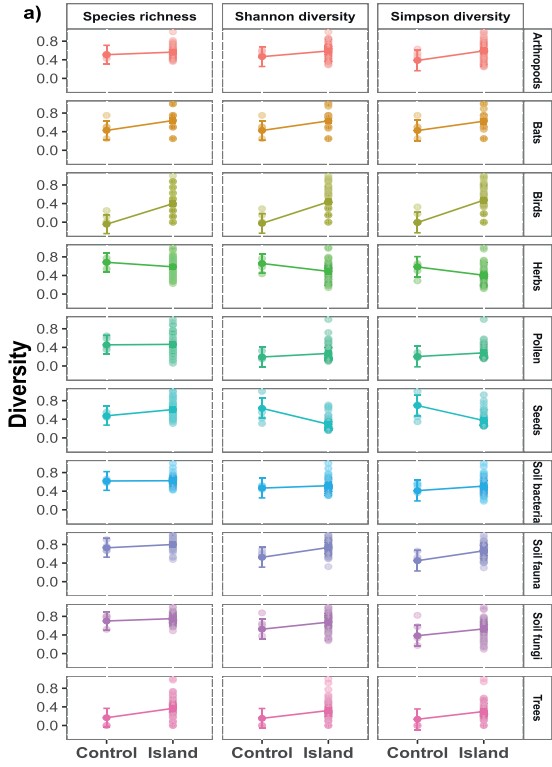

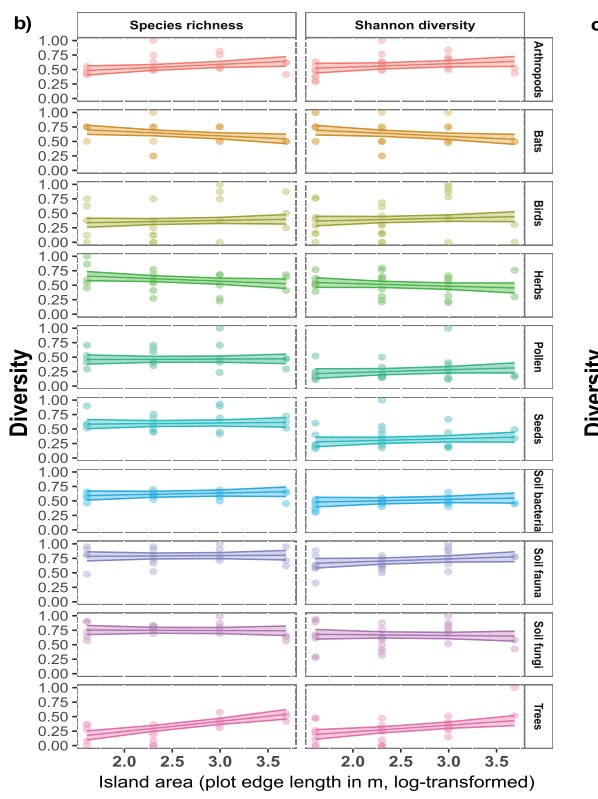

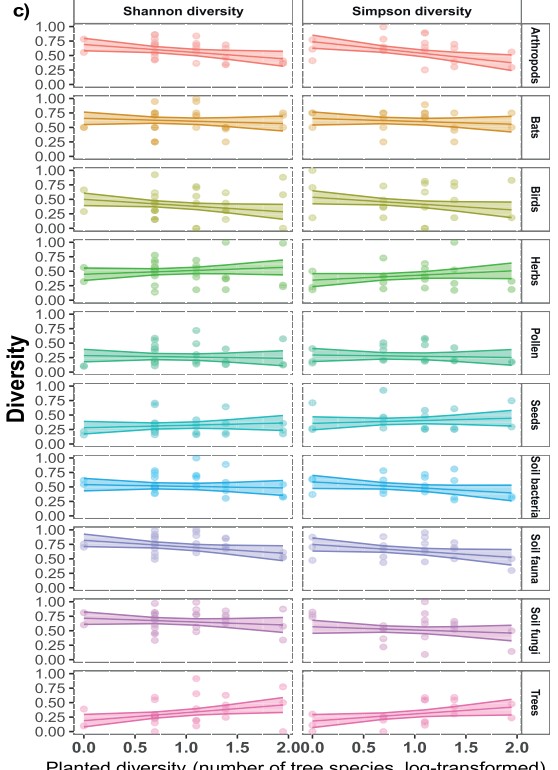

**Extended Data Fig. 1 | Interaction between the experimental treatment (island, area, planted diversity) and biodiversity indicator.** (**a**) Interaction island × indicator. (**b**) Interaction area × indicator. (**c**) Interaction planted diversity × indicator. Lines/Solid points are linear mixed-effect model fits using ggeffect: (a) centre for the error bars indicate marginal means (b-c) bands indicate 95% confidence intervals with the centre line indicating marginal means using ggeffect. Points are observed values (n = 56 study plots). Only significant interaction terms in the analysis of variance of the linear mixed-effect models are shown (p < 0.05).

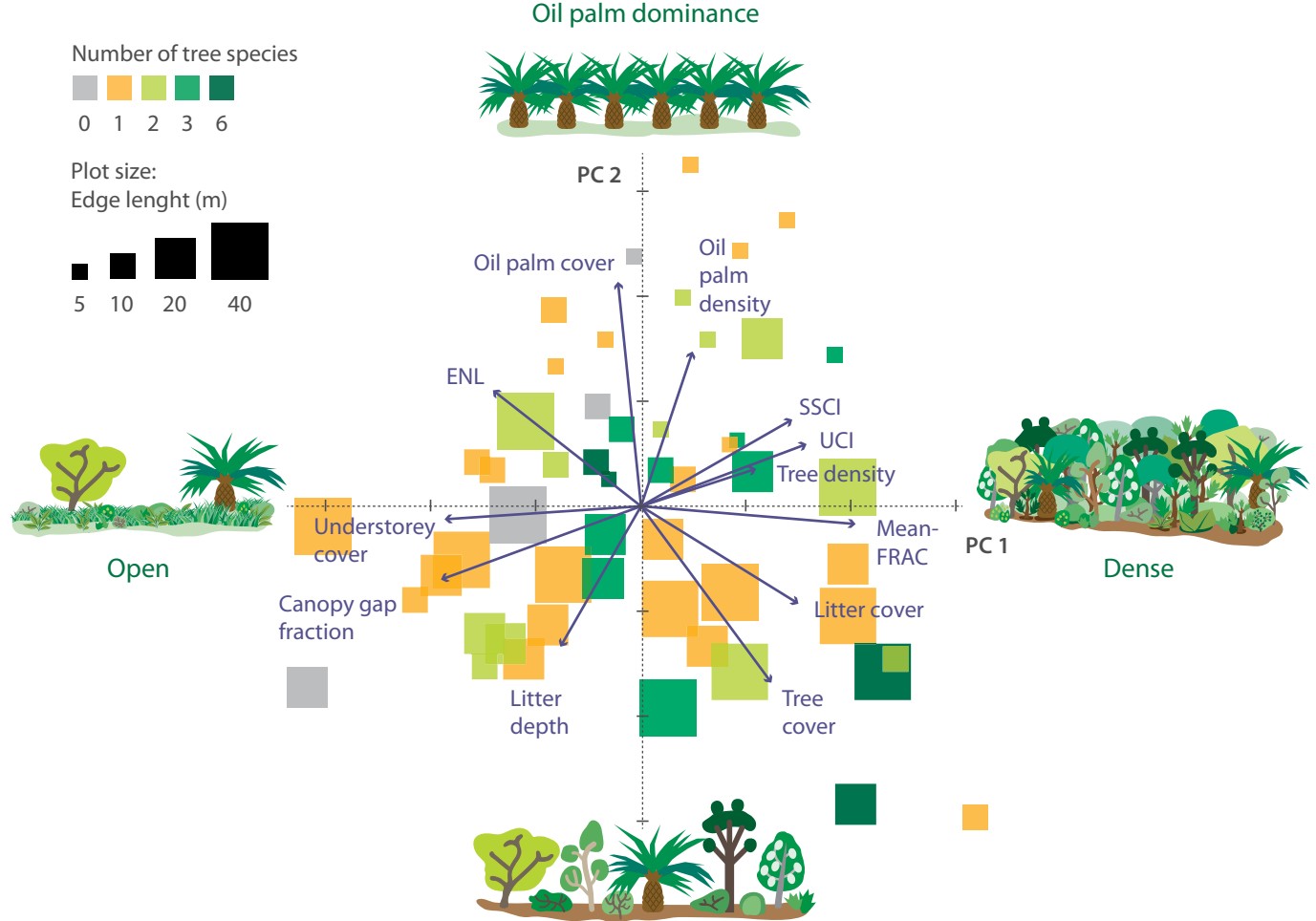

**Extended Data Fig. 2 | Principal components of the vegetation structure in the tree islands.** Each square represents one of the 52 tree islands, which vary in planted tree diversity (colour) and area (size of the square). The first two components (PC1 and PC2) explain 32% and 21% of the total variance, respectively. Vegetation structure variables included (variable loadings PC1 and PC2): Oil palm cover (−0.13, 1.26), ENL, i.e., effective number of layers (−0.83, 0.65), understorey cover (−1.10, −0.07), canopy gap fraction (−1.12, −0.41), litter depth (−0.45, −0.79), tree cover (0.73, −0.99), litter cover (0.88, −0.55), MeanFRAC, i.e., a measure of the geometric complexity of the vegetation structure that is density-dependent (1.20, −0.10), tree density (0.64, 0.23), UCI, i.e., understorey complexity index (0.93, 0.35), SSCI, i.e., stand structural complexity index (0.85, 0.49) and oil palm density (0.64, 0.23). PC1 is mostly associated with dense and structurally complex vegetation and with low understorey vegetation cover (see Methods for details). PC2 is mostly associated with a high proportion of palms and low proportion of trees in the canopy. In the main text, we refer to PC1 as 'structural complexity' and minus PC2 as 'tree dominance'.

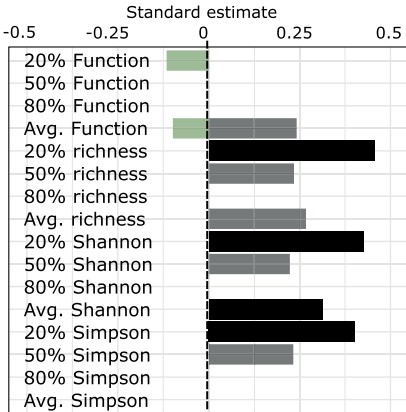

**Extended Data Fig. 3 | Direct and indirect influence of tree island size and planted tree diversity on multidiversity and multifunctionality.** Results of the structural equation models (i.e. standard coefficient estimates) for multifunctionality and multidiversity (based on species richness, Shannon diversity and Simpson diversity) with alternative calculation methods (average or thresholding approach - see Methods). The effect of tree island area can be direct (black bars) or *via* tree dominance (grey bars), and the effect of planted tree diversity can be direct (dark green bars - here absent) or via structural complexity (light green bars). Only significant estimates are presented (p < 0.05 two-sided ANOVA, n = 52 tree islands).

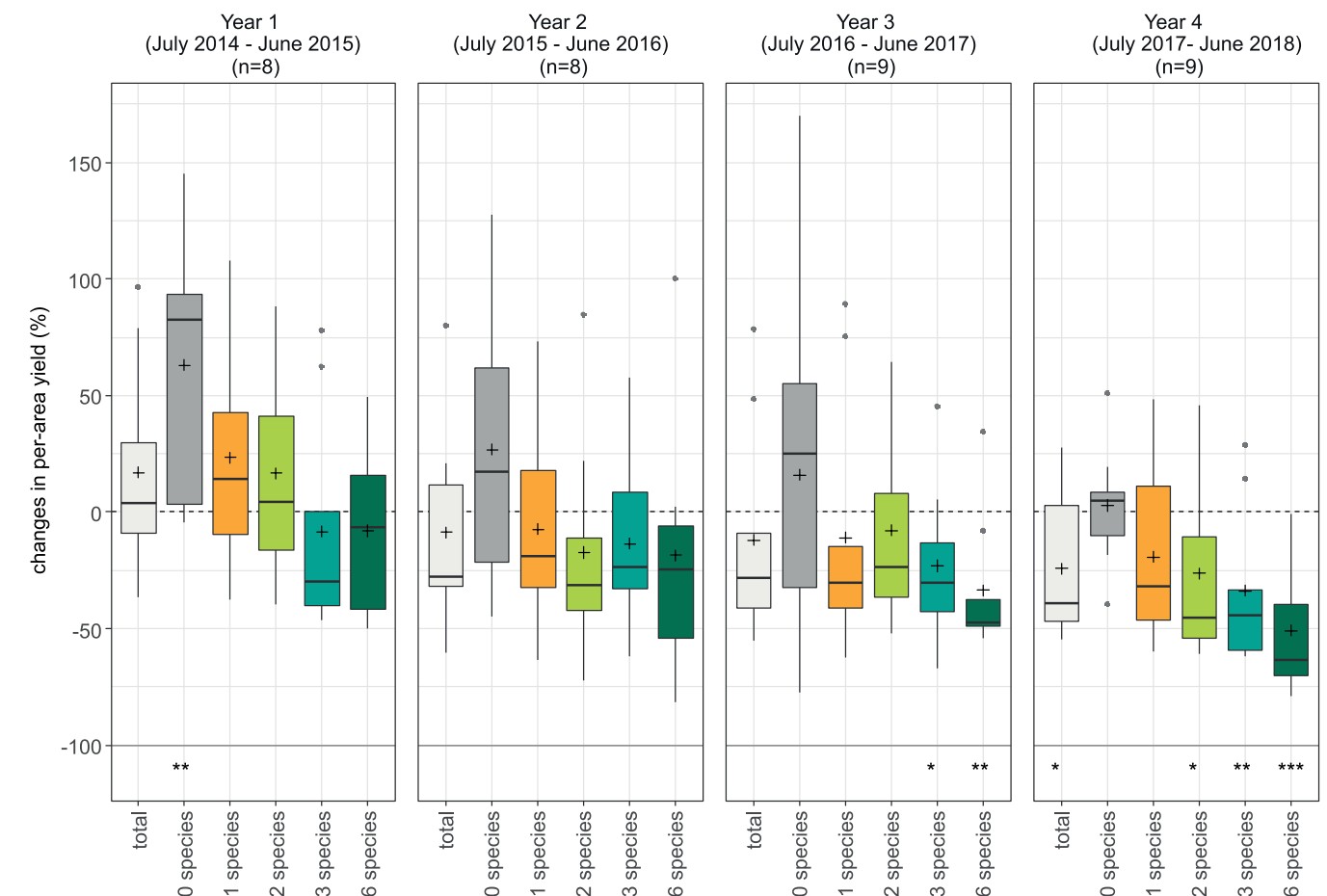

**Extended Data Fig. 4 | Changes in per area yield across four years for different planted diversity levels.** Relative changes in per area yield ($\Delta Y_{ha}/Y_{ha\_ref}$ in %) compared to conventionally managed oil palm monocultures ('reference') for different planted tree species diversity: 0, 1, 2, 3, 6 species, and all 52 tree islands combined ("total"). Plus signs represent means, horizontal lines medians, boxes interquartile ranges and vertical bars ranges. The dashed horizontal line is the mean of the reference palms (i.e. the palms in adjacent position 3). 'n'-values denote the number of months included in the annual boxes. Significant differences to reference as based on Mann-Whitney tests are indicated at the bottom of each panel, with the significance levels * ($p < 0.1$), ** ($p < 0.05$) and *** ($p < 0.01$).

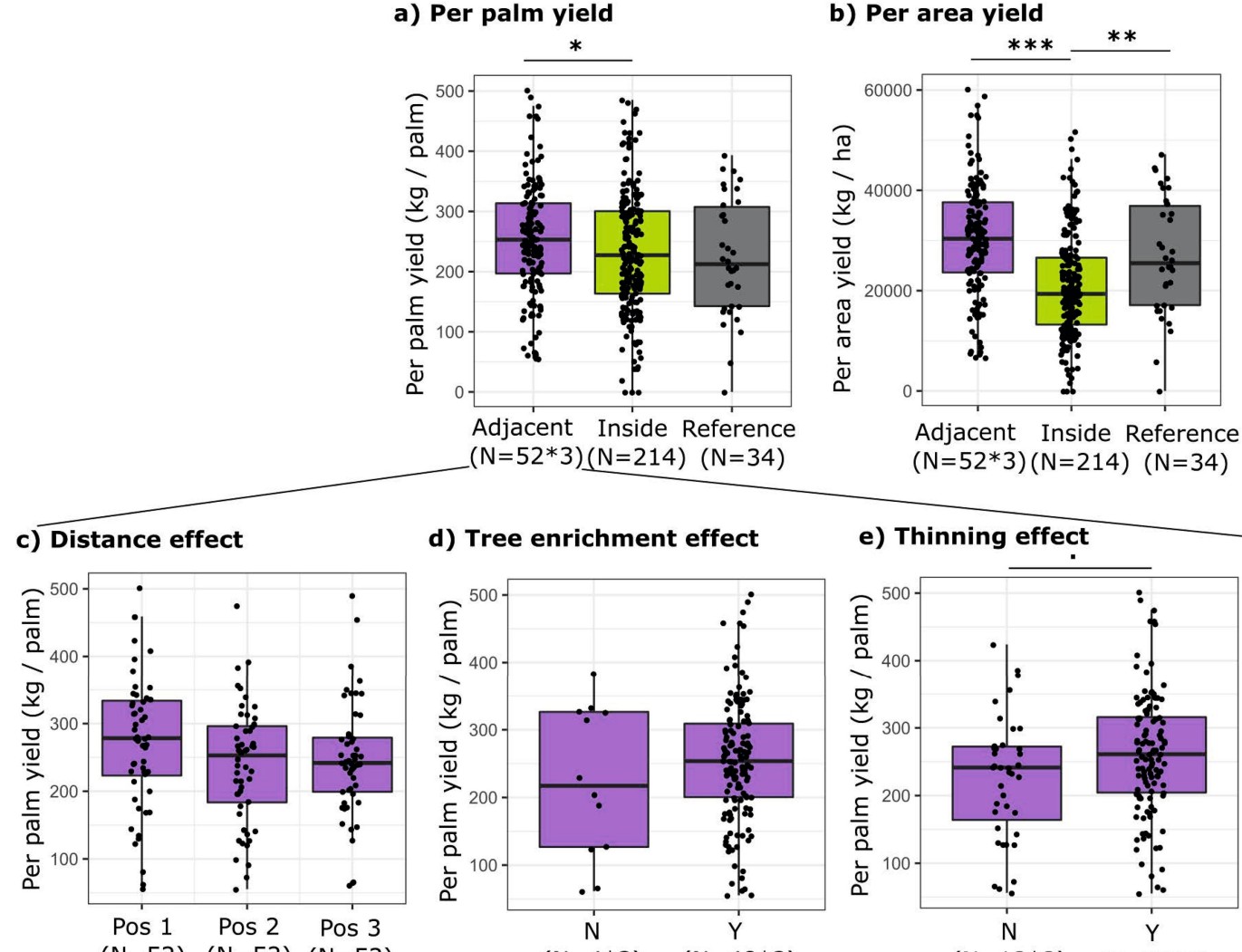

**Extended Data Fig. 5 | Effects of the experimental treatment on per palm and per area yield.** Each dot represents an individual oil palm that was monitored within our study (N = 404 in total, from October 2017 to November 2018). In each boxplot, the horizontal line corresponds to the median; the lower and upper hinges correspond to the first and third quartiles and the upper and lower whiskers are limited by the 1.5 the interquartile ranges. The overall effects of the experimental treatment on (**a**) per palm yield (in kg/palm) and (**b**) per area yield (in kg/ha) were significant in both cases ($\chi^2 = 6.39$, df = 2, p-value = 0.04 and $\chi^2 = 75.35$, df = 2, p-value < 0.001, respectively). Multi-comparison indicates that (**a**) per palm yield inside the islands was lower than per palm yield adjacent to the tree islands; (**b**) per area yield inside the tree islands were lower than per area yield adjacent to the tree islands and per area

yield in the conventionally managed oil palm monocultures. **c**) There was no effect of the distance to the tree island edge (i.e. positions 1, 2 and 3) on the per palm yield of the adjacent oil palms ($\chi^2 = 4.47$, df = 2, p-value = 0.10). **d**) There was no effect of tree planting on the per palm yield of the adjacent oil palms ($\chi^2 = 1.02$, df = 1, p-value = 0.31). **e**) The effect of oil palm thinning on per palm yield of the adjacent palms was marginally significant ($\chi^2 = 3.78$, df = 1, p-value = 0.05). Multi-comparisons were conducted with post hoc Tukey tests. More details of the statistical analysis are provided in section 2.3.2 of[19], where models number one, two, three and four correspond here to the panels a), b), c) and e), respectively. Significance levels are indicated by. (p < 0.1), * (p < 0.05), ** (p < 0.01) and *** (p < 0.001).

**Extended Data Table 1 | ANOVA of the linear mixed-effect models for biodiversity and ecosystem functioning**

| Explanatory variables | numDF | denDF | Biodiversity Indicators | | | | | | Ecosystem functioning Indicators | | | |
| | | | Species richness | | Shannon diversity | | Simpson diversity | | | | | |
| | | | F-value | p-value | F-value | p-value | F-value | p-value | numDF | denDF | F-value | p-value |
| Intercept | 1 | 459 | 5437,8 | **<0.0001** | 3519,8 | **<0.0001** | 2887,8 | **<0.0001** | 1 | 918 | 4233,8 | **<0.0001** |
| Tree island | 1 | 51 | 10,793 | **0,0018** | 4,579 | **0,0372** | 4,716 | **0,0346** | 1 | 51 | 6,233 | **0,0158** |
| Island area | 1 | 51 | 3,793 | *0,057* | 5,798 | **0,0197** | 3,975 | *0,0515* | 1 | 51 | 12,851 | **0,0008** |
| Planted diversity | 1 | 51 | 0,600 | 0,4423 | 1,976 | 0,1658 | 3,73 | *0,0590* | 1 | 51 | 0,216 | 0,6438 |
| Indicator | 9 | 459 | 37,994 | **<0.0001** | 42,321 | **<0.0001** | 24,546 | **<0.0001** | 18 | 918 | 95,173 | **<0.0001** |
| Island area × planted diversity | 1 | 51 | 0,201 | 0,6554 | 1,465 | 0,2318 | 0,852 | 0,3603 | 1 | 51 | 2,194 | 0,1447 |
| Tree island × indicator | 9 | 459 | 2,545 | **0,0074** | 3,589 | **0,0002** | 3,047 | **0,0015** | 18 | 918 | 0,918 | 0,5564 |
| Island area × indicator | 9 | 459 | 5,059 | **<0.0001** | 2,805 | **0,0033** | 1,836 | 0,0598 | 18 | 918 | 1,382 | 0,1319 |
| Planted diversity × indicator | 9 | 459 | 1,310 | 0,2291 | 2,326 | **0,0144** | 2,768 | **0,0037** | 18 | 918 | 0,987 | 0,4718 |
| Island area × planted diversity × indicator | 9 | 459 | 1,453 | 0,1628 | 1,380 | 0,1945 | 1,164 | 0,3166 | 18 | 918 | 0,930 | 0,5410 |

Each biodiversity and ecosystem functioning indicator was standardized to unit scale. Plot was included as a random term. Two-sided ANOVA (n=56 study plots). numDF: degrees of freedom, denDF: denominator degrees of freedom.

**Extended Data Table 2 | Summary statistics of piecewise structural equation models for each biodiversity and ecosystem functioning indicators and multidiversity and multifunctionality**

| | Indicator | Fisher C | df | p-value | AIC | AICc | BIC | K | n | R² |
|---|---|---|---|---|---|---|---|---|---|---|
| Species richness | Arthropods | 5.739 | 6 | 0.453 | 29.739 | 39.652 | 53.154 | 12 | 52 | 0.52 |
| | Bats | 12.265 | 10 | 0.268 | 32.265 | 40.921 | 51.777 | 10 | 52 | 0.08 |
| | Birds | 11.075 | 10 | 0.352 | 31.075 | 39.412 | 50.587 | 10 | 52 | 0.04 |
| | Herbs | 12.653 | 10 | 0.244 | 32.653 | 41.414 | 52.165 | 10 | 52 | 0.08 |
| | Pollen | 7.998 | 10 | 0.629 | 27.998 | 35.51 | 47.51 | 10 | 52 | 0.01 |
| | Seeds | 5.739 | 6 | 0.453 | 29.739 | 39.652 | 53.154 | 12 | 52 | 0.30 |
| | Trees | 9.799 | 10 | 0.458 | 29.799 | 37.794 | 49.311 | 10 | 52 | 0.43 |
| | Soil fauna | 10.078 | 10 | 0.434 | 30.078 | 38.148 | 49.59 | 10 | 52 | 0.01 |
| | Soil bacteria | 7.358 | 10 | 0.691 | 27.358 | 34.698 | 46.87 | 10 | 52 | 0.05 |
| | Soil fungi | 7.592 | 10 | 0.669 | 27.592 | 34.995 | 47.104 | 10 | 52 | 0.00 |
| | Average multidiversity | 5.794 | 8 | 0.67 | 27.794 | 36.132 | 49.258 | 11 | 52 | 0.19 |
| | 20% threshold multidiversity | 10.827 | 10 | 0.371 | 30.827 | 39.098 | 50.339 | 10 | 52 | 0.21 |
| | 50% threshold multidiversity | 5.939 | 8 | 0.654 | 27.939 | 36.321 | 49.403 | 11 | 52 | 0.17 |
| | 80% threshold multidiversity | 11.614 | 10 | 0.312 | 31.614 | 40.096 | 51.126 | 10 | 52 | 0.02 |
| Shannon diversity | Arthropods | 15.499 | 10 | 0.115 | 35.499 | 45.023 | 55.011 | 10 | 52 | 0.19 |
| | Bats | 6.334 | 8 | 0.61 | 28.334 | 36.834 | 49.798 | 11 | 52 | 0.16 |
| | Birds | 9.966 | 10 | 0.443 | 29.966 | 38.006 | 49.478 | 10 | 52 | 0.05 |
| | Herbs | 8.089 | 8 | 0.425 | 30.089 | 39.116 | 51.553 | 11 | 52 | 0.15 |
| | Pollen | 8.066 | 10 | 0.622 | 28.066 | 35.596 | 47.578 | 10 | 52 | 0.02 |
| | Seeds | 8.071 | 10 | 0.622 | 28.071 | 35.602 | 47.583 | 10 | 52 | 0.07 |
| | Trees | 14.036 | 10 | 0.171 | 34.036 | 43.168 | 53.548 | 10 | 52 | 0.29 |
| | Soil fauna | 6.652 | 10 | 0.758 | 26.652 | 33.803 | 46.164 | 10 | 52 | 0.2 |
| | Soil bacteria | 6.298 | 10 | 0.79 | 26.298 | 33.354 | 45.81 | 10 | 52 | 0.07 |
| | Soil fungi | 6.191 | 10 | 0.799 | 26.191 | 33.218 | 45.703 | 10 | 52 | 0.02 |
| | Average multidiversity | 13.088 | 10 | 0.219 | 33.088 | 41.965 | 52.6 | 10 | 52 | 0.13 |
| | 20% threshold multidiversity | 7.082 | 10 | 0.718 | 27.082 | 34.348 | 46.594 | 10 | 52 | 0.21 |
| | 50% threshold multidiversity | 5.784 | 8 | 0.671 | 27.784 | 36.119 | 49.248 | 11 | 52 | 0.10 |
| | 80% threshold multidiversity | 8.444 | 10 | 0.586 | 28.444 | 36.075 | 47.956 | 10 | 52 | 0.02 |
| Simpson diversity | Arthropods | 13.674 | 10 | 0.188 | 33.674 | 42.708 | 53.186 | 10 | 52 | 0.21 |
| | Bats | 6.427 | 8 | 0.6 | 28.427 | 36.955 | 49.891 | 11 | 52 | 0.17 |
| | Birds | 9.441 | 10 | 0.491 | 29.441 | 37.34 | 48.953 | 10 | 52 | 0.05 |
| | Herbs | 7.394 | 8 | 0.495 | 29.394 | 38.212 | 50.858 | 11 | 52 | 0.13 |
| | Pollen | 9.115 | 10 | 0.521 | 29.115 | 36.926 | 48.627 | 10 | 52 | 0.04 |
| | Seeds | 7.58 | 10 | 0.67 | 27.58 | 34.98 | 47.092 | 10 | 52 | 0.07 |
| | Trees | 10.082 | 8 | 0.259 | 32.082 | 41.707 | 53.546 | 11 | 52 | 0.28 |
| | Soil fauna | 7.177 | 10 | 0.709 | 27.177 | 34.468 | 46.689 | 10 | 52 | 0.19 |
| | Soil bacteria | 6.95 | 10 | 0.73 | 26.95 | 34.18 | 46.462 | 10 | 52 | 0.11 |
| | Soil fungi | 5.841 | 10 | 0.828 | 25.841 | 32.774 | 45.353 | 10 | 52 | 0.01 |
| | Average multidiversity | 11.432 | 10 | 0.325 | 31.432 | 39.865 | 50.944 | 10 | 52 | 0.13 |
| | 20% threshold multidiversity | 8.472 | 10 | 0.583 | 28.472 | 36.111 | 47.984 | 10 | 52 | 0.21 |
| | 50% threshold multidiversity | 6.094 | 8 | 0.637 | 28.094 | 36.522 | 49.558 | 11 | 52 | 0.12 |
| | 80% threshold multidiversity | 9.598 | 10 | 0.476 | 29.598 | 37.539 | 49.11 | 10 | 52 | 0.04 |
| Ecosystem functioning | Oil palm yield | 9.408 | 8 | 0.309 | 31.408 | 40.83 | 52.872 | 11 | 52 | 0.13 |
| | Microclimate buffering | 7.979 | 8 | 0.436 | 29.979 | 38.973 | 51.443 | 11 | 52 | 0.53 |
| | Water infiltration | 7.606 | 8 | 0.473 | 29.606 | 38.488 | 51.07 | 11 | 52 | 0.21 |
| | Evapotranspiration | 11.025 | 10 | 0.356 | 31.025 | 39.349 | 50.537 | 10 | 52 | 0.10 |
| | Litter input | 13.063 | 8 | 0.11 | 35.063 | 45.582 | 56.527 | 11 | 52 | 0.48 |
| | Litter decomposition | 9.598 | 10 | 0.476 | 29.598 | 37.539 | 49.11 | 10 | 52 | 0.00 |
| | Decomposers | 11.613 | 10 | 0.312 | 31.613 | 40.095 | 51.125 | 10 | 52 | 0.00 |
| | Herbivores | 12.385 | 10 | 0.26 | 32.385 | 41.074 | 51.897 | 10 | 52 | 0.04 |
| | Predators (soil fauna) | 8.438 | 10 | 0.586 | 28.438 | 36.068 | 47.95 | 10 | 52 | 0.00 |
| | Predators (arthropods) | 6.445 | 10 | 0.777 | 26.445 | 33.54 | 45.957 | 10 | 52 | 0.05 |
| | Predators (vertebrates) | 11.575 | 8 | 0.171 | 33.575 | 43.648 | 55.039 | 11 | 52 | 0.21 |
| | 1 / soil C:N | 12.551 | 10 | 0.25 | 32.551 | 41.284 | 52.063 | 10 | 52 | 0.03 |
| | Soil decompaction | 6.081 | 8 | 0.638 | 28.081 | 36.505 | 49.545 | 11 | 52 | 0.21 |
| | Soil P | 9.188 | 10 | 0.514 | 29.188 | 37.019 | 48.7 | 10 | 52 | 0.15 |
| | Pollination rate | 6.841 | 10 | 0.74 | 26.841 | 34.042 | 46.353 | 10 | 52 | 0.01 |
| | Pollinators | 5.739 | 6 | 0.453 | 29.739 | 39.652 | 53.154 | 12 | 52 | 0.38 |
| | Resistance to invasive plants | 11.203 | 10 | 0.342 | 31.203 | 39.575 | 50.715 | 10 | 52 | 0.02 |
| | Native seeds | 8.87 | 10 | 0.545 | 28.87 | 36.616 | 48.382 | 10 | 52 | 0.03 |
| | Aboveground biomass | 10.932 | 8 | 0.206 | 32.932 | 42.812 | 54.396 | 11 | 52 | 0.21 |
| | Average multifunctionality | 5.739 | 6 | 0.453 | 29.739 | 39.652 | 53.154 | 12 | 52 | 0.37 |
| | 20% threshold multifunctionality | 6.041 | 8 | 0.643 | 28.041 | 36.453 | 49.505 | 11 | 52 | 0.18 |
| | 50% threshold multifunctionality | 11.113 | 10 | 0.349 | 31.113 | 39.46 | 50.625 | 10 | 52 | 0.08 |
| | 80% threshold multifunctionality | 10.852 | 10 | 0.369 | 30.852 | 39.129 | 50.364 | 10 | 52 | 0.06 |

Every row is an individual model. Fisher's C statistic, degrees of freedom (df), significance value based on a Chi-square test, Information criterion (Akaike, corrected Akaike and Bayesian), likelihood degrees of freedom (K), sample size (n=52 tree islands) and R-squared values.

**Extended Data Table 3 | Result of the principal component analysis of the vegetation structure variables.**
**(n = 52 tree islands)**

| Vegetation structure | PC1 | PC2 | PC3 | PC4 | PC5 | PC6 |
|---|---|---|---|---|---|---|
| Litter cover | 0.88 | -0.55 | 0.77 | -0.13 | 0.31 | -0.04 |
| SSCI | 0.85 | 0.49 | -0.85 | -0.26 | 0.36 | 0.26 |
| ENL | -0.83 | 0.65 | -0.11 | -0.65 | 0.13 | 0.22 |
| UCI | 0.93 | 0.35 | -0.62 | 0.05 | -0.15 | -0.33 |
| MeanFRAC | 1.2 | -0.1 | -0.63 | 0.16 | 0.22 | 0.08 |
| Understory cover | -1.1 | -0.07 | -0.71 | 0.06 | -0.27 | -0.15 |
| Litter depth | -0.45 | -0.79 | -0.28 | 0.15 | 0.65 | -0.78 |
| Tree cover | 0.73 | -0.99 | -0.22 | -0.18 | -0.46 | 0.08 |
| Oil palm cover | -0.13 | 1.26 | 0.21 | 0.04 | 0.42 | -0.2 |
| Oil palm density | 0.29 | 0.86 | 0.13 | 0.57 | -0.57 | -0.41 |
| Tree density | 0.64 | 0.23 | 0.19 | -1.01 | -0.29 | -0.57 |
| Canopy gap fraction | -1.12 | -0.41 | -0.36 | -0.29 | -0.13 | -0.07 |

**Extended Data Table 4 | List of soil fauna groups and associated guild**

| Group | Guild |
|---|---|
| Annelida | Decomposers |
| Araneae | Predators |
| Blattodea | Decomposers |
| Chilopoda | Predators |
| Coleoptera - Carabidae and Staphylinidae | Predators |
| Coleoptera - Curculionidae | Herbivores |
| Coleoptera - Others | Decomposers |
| Dermaptera | Decomposers |
| Diplopoda | Decomposers |
| Diplura - Campodeidae | Decomposers |
| Diplura - Japygidae | Predators |
| Diptera | Decomposers |
| Formicidae | Ants |
| Hemiptera - Auchenorrhyncha and Sternorrhyncha | Herbivores |
| Heteroptera | Predators |
| Hymenoptera | Predators |
| Isopoda | Decomposers |
| Isoptera | Decomposers |
| Lepidoptera | Herbivores |
| Mesostigmata | Predators |
| Opiliones | Predators |
| Oribatida | Decomposers |
| Orthoptera | Herbivores |
| Pauropoda | Decomposers |
| Prostigmata | Predators |
| Protura | Decomposers |
| Pseudoscorpiones | Predators |
| Psocoptera | Decomposers |
| Schizomida | Predators |
| Symphyla | Decomposers |
| Thysanoptera | Herbivores |

# Reporting Summary

## Statistics

For all statistical analyses, confirm that the following items are present in the figure legend, table legend, main text, or Methods section.

| n/a | Confirmed | |
|---|---|---|
| ☐ | ☒ | The exact sample size (*n*) for each experimental group/condition, given as a discrete number and unit of measurement |
| ☐ | ☒ | A statement on whether measurements were taken from distinct samples or whether the same sample was measured repeatedly |
| ☐ | ☒ | The statistical test(s) used AND whether they are one- or two-sided<br>*Only common tests should be described solely by name; describe more complex techniques in the Methods section.* |
| ☐ | ☒ | A description of all covariates tested |
| ☐ | ☒ | A description of any assumptions or corrections, such as tests of normality and adjustment for multiple comparisons |
| ☐ | ☒ | A full description of the statistical parameters including central tendency (e.g. means) or other basic estimates (e.g. regression coefficient) AND variation (e.g. standard deviation) or associated estimates of uncertainty (e.g. confidence intervals) |
| ☐ | ☒ | For null hypothesis testing, the test statistic (e.g. *F*, *t*, *r*) with confidence intervals, effect sizes, degrees of freedom and *P* value noted<br>*Give P values as exact values whenever suitable.* |
| ☐ | ☒ | For Bayesian analysis, information on the choice of priors and Markov chain Monte Carlo settings |
| ☐ | ☒ | For hierarchical and complex designs, identification of the appropriate level for tests and full reporting of outcomes |
| ☐ | ☒ | Estimates of effect sizes (e.g. Cohen's *d*, Pearson's *r*), indicating how they were calculated |

*Our web collection on statistics for biologists contains articles on many of the points above.*

## Software and code

Policy information about availability of computer code

| Data collection | The data were processed and analyzed in R version 1.2.1335 |
|---|---|
| Data analysis | Data and R code are available at https://doi.org/10.6084/m9.figshare.22320490 |

For manuscripts utilizing custom algorithms or software that are central to the research but not yet described in published literature, software must be made available to editors and reviewers. We strongly encourage code deposition in a community repository (e.g. GitHub). See the Nature Portfolio guidelines for submitting code & software for further information.

## Data

Policy information about availability of data

All manuscripts must include a data availability statement. This statement should provide the following information, where applicable:
- Accession codes, unique identifiers, or web links for publicly available datasets
- A description of any restrictions on data availability
- For clinical datasets or third party data, please ensure that the statement adheres to our policy

The raw data are available at https://data.goettingen-research-online.de/dataverse/crc990, with the specific link for each dataset provided in the Supplementary Tables 1 - 3. The processed data is available at https://doi.org/10.6084/m9.figshare.22320490.
Seed DNA sequences are available in NCBI Genbank under the accession numbers HYPERLINK "https://www.ncbi.nlm.nih.gov/nuccore/OM811991.1/" OM811991- HYPERLINK "https://www.ncbi.nlm.nih.gov/nuccore/OM812021" OM812021, HYPERLINK "https://www.ncbi.nlm.nih.gov/nuccore/OM837673" OM837673- HYPERLINK "https://www.ncbi.nlm.nih.gov/nuccore/OM837724" OM837724, and HYPERLINK "https://www.ncbi.nlm.nih.gov/nuccore/OM935782" OM935782- HYPERLINK "https://www.ncbi.nlm.nih.gov/nuccore/OM935815" OM935815. Sequencing data of the soil fungal community were deposited in the NCBI Sequence Read Archive (SRA) under Bioproject accession number PRJNA659225. The public UNITE database (https://unite.ut.ee/) v7.2 on fungal ITS sequences was used as a reference of taxonomic classification. Sequence data of the bacterial communities were deposited in the NCBI SRA under Bioproject accession number PRJNA841353. Sequence identification was performed by mapping all curated sequences against the SILVA database version 132 (https://www.arb-silva.de/).

## Human research participants

Policy information about studies involving human research participants and Sex and Gender in Research.

| | |
|---|---|
| Reporting on sex and gender | *N I A* |
| Population characteristics | *N I A* |
| Recruitment | *N I A* |
| Ethics oversight | N/A |

Note that full information on the approval of the study protocol must also be provided in the manuscript.

# Field-specific reporting

Please select the one below that is the best fit for your research. If you are not sure, read the appropriate sections before making your selection.

☐ Life sciences   ☐ Behavioural & social sciences   ☒ Ecological, evolutionary & environmental sciences

For a reference copy of the document with all sections, see nature.com/documents/nr-reporting-summary-flat.pdf

# Ecological, evolutionary & environmental sciences study design

All studies must disclose on these points even when the disclosure is negative.

**Study description**

Our study was conducted in EFForTS-BEE, the Biodiversity Enrichment Experiment of the EFForTS project [Ecological and Socioeconomic Functions of Tropical Lowland Rainforest Transformation Systems (Sumatra, Indonesia)]. EFForTS-BEE is part of the global network of tree diversity experiments TreeDivNet (https://treedivnet.ugent.be/). The study region is characterized by a humid tropical climate with a mean temperature of 26.7 ± 0.2°C and an annual rainfall of 2,235 ± 381 mm and the dominant soil type is loamy Acrisol. In December 2013, 52 experimental plots (i.e. tree islands) were established in a conventional 140 ha oil palm plantation. Following a random partition design, we systematically varied plot area (25, 100, 400 and 1600 m2) and tree species diversity (0, 1, 2, 3 and 6 species). The six planted tree species (Archidendron jiringa (Jack) I.C.Nielsen. (Fabaceae), Parkia speciosa Hassk. (Fabaceae), Durio zibethinus L. (Malvaceae), Dyera polyphylla (Miq.) Steenis (Apocynaceae), Shorea leprosula Miq. (Dipterocarpaceae) and Peronema canescens Jack (Lamiaceae)) are native to the region and widely used for their fruits, timber or latex. Around 40% of the oil palms located inside the tree islands were felled, with the number of felled oil palms differed depending on the tree island area. The trees were planted between the felled and standing oil palms on a 2-m triangular grid. The tree islands were fenced, and the management comprised a total stop of fertilizer, herbicide and pesticide application after planting. After May 2016, manual weeding was restricted to 1-m circles around the planted trees when these were shorter than the surrounding grass layer, allowing for natural regeneration. In addition to the 52 tree islands, we established four control plots in the conventional oil palm plantation that were managed as usual, in the main text referred to as conventional monocultures. In total, the study comprised 56 plots. In each study plot larger than 25 m2, one subplot of 5 m x 5 m was established in a random location at a minimum distance of 1.5 m from the plot edge.

**Research sample**

Ecosystem functioning
We measured 20 variables related to seven categories of ecosystem functioning including Productivity: (0) tree growth (basal area increment of the planted trees in m2/ha/year) that was further excluded from the analysis - see method on multifunctionality, (1) Oil palm yield (per island oil palm yield changes in kg of fresh fruit bunches / island), (2) aboveground biomass (biomass stored in the aerial parts of the planted trees and the oil palms, in t/ha), Resistance to invasion: (3) native seeds (total number of arriving native seeds / m2); (4) resistance to invasive plants (100 – observed cover of Clidemia hirta, in %); Pollination: (5) pollinators (number of sampled individuals), (6) pollination rate (fraction of flowers on phytometer plants that are pollinated, %), Soil quality: (7) soil P (phosphorous content, %), (8) 1 / soil C:N (that is the molar ratio of soil C content to soil N content), (9) soil decompaction (inverse of soil bulk density in g/cm3); Predation and herbivory: (10) predatory invertebrates (total activity duration of insectivorous bats and birds, in seconds); (11) predatory arthropods (number of sampled individuals), (12) predatory soil fauna (energy flux, in J/hour), (13) herbivory (energy flux, in J/hour); Carbon & nutrient cycling: (14) Decomposers (energy flux, in J/hour); (15) litter decomposition (relative biomass loss of litter after 6 months in litterbags, %), (16) litter input (biomass of leaf litter falling in traps, g / m2), water and climate regulation: (17) evapotranspiration (canopy latent heat flux, in W/m2); (18) soil water infiltration capacity (saturated soil

hydraulic conductivity in cm/h), (19) micro-climate buffering (median daily amplitude of air temperature during one year, °C / day). A more detailed summary of the 20 ecosystem functioning variables is presented in Extended Data Table 1.
Biodiversity
We derived taxonomic diversity for soil bacteria and soil fungi, soil fauna, herbs, trees, seeds, pollen, understory arthropods, birds and bats. Most of the groups (arthropods, herbs, trees, birds, seeds) were sorted at the lowest possible taxonomic level (species or morphospecies). Pollen, soil fauna and bats were sorted to higher levels, mainly family, order and morphotypes, respectively. Soil bacteria and soil fungi were analyzed by DNA based marker gene sequencing as amplicons sequence variants (ASVs) or operational taxonomic units (OTU), respectively. Hereafter we refer to these different taxonomic units (species, family, order, morphotypes and OTU) as 'species' for simplicity.

**Sampling strategy**

In all the 56 study plots, multiple indicators related to biodiversity, ecosystem functioning, and structure were measured using standardized procedures and constant sampling areas at the level of the plot (i.e. tree island) or subplot (see Extended Data Tables 1-3). Only trees were sampled at unequal areas (i.e. all trees present in the plots were sampled) and were therefore standardized using rarefaction curves.

**Data collection**

In all the 56 study plots, multiple indicators related to biodiversity, ecosystem functioning, and structure were measured using standardized procedures and constant sampling areas at the level of the plot (i.e. tree island) or subplot (see Extended Data Tables 1-3).

**Timing and spatial scale**

We conducted an interdisciplinary field campaign from October 2016 to October 2018, i.e. 33 to 57 months after establishment of the experiment. At this early stage of the experiment, the tree islands already differed in their structural complexity and the planted trees reached up to 16 m height.

**Data exclusions**

As tree growth and leaf litter input were correlated and formed cluster, we excluded tree growth from the analysis. For leaf litter input, we then excluded outliers defined as plot-level values outside the range of 3 standard deviations around the median of the entire data (less than 5% of the litterweight data, total and per species).

**Reproducibility**

The Biodiversity Enrichment Experiment in Oil Palm Plantations (EFForTS-BEE) is unique but can be replicated to other landscapes. All data and code to reproduce the analyses are included at https://doi.org/10.6084/m9.figshare.22320490

**Randomization**

Each variable presented in the main text had one measurement per plot, such that randomization was not applicable.

**Blinding**

Each variable presented in the main text had one measurement per plot, such that blinding was not applicable.

**Did the study involve field work?** ☒ Yes ☐ No

# Field work, collection and transport

**Field conditions**

The study region is characterized by a humid tropical climate with a mean temperature of 26.7 ± 0.2°C and an annual rainfall of 2,235 ± 381 mm and the dominant soil type is loamy Acrisol.

**Location**

The experiment was established on an oil- palm plantation of PT. Humusindo Makmur Sejati (01.95∘ S and 103.25∘ E, 47 ± 11 m a.s.l.) near Bungku village in the lowlands of Jambi province, Sumatra. The specific longitude and latitude for each plots are included in: https://doi.org/10.6084/m9.figshare.22320490

**Access & import/export**

Research permits were granted by the Indonesia Ministry of Research and Technology (Ristek or Ristekdikti) for all researchers involved in data collection for this study.
For soil fungi: The ITS2 sequences were deposited in the National Center for Biotechnology Information (NCBI) Sequence Read Archive (SRA) under bioproject accession number PRJNA659225.
For soil bacteria: The 16S rRNA gene and transcript sequences were deposited in the National Center for Biotechnology Information (NCBI) Sequence Read Archive (SRA) under bioproject accession number PRJNA841353

**Disturbance**

Major disturbance occured during the establishment of the experiment three to five years prior to the study presented here.

# Reporting for specific materials, systems and methods

We require information from authors about some types of materials, experimental systems and methods used in many studies. Here, indicate whether each material, system or method listed is relevant to your study. If you are not sure if a list item applies to your research, read the appropriate section before selecting a response.

## Materials & experimental systems

| n/a | Involved in the study |
|---|---|
| ☒ | ☐ Antibodies |
| ☒ | ☐ Eukaryotic cell lines |
| ☒ | ☐ Palaeontology and archaeology |
| ☐ | ☒ Animals and other organisms |
| ☒ | ☐ Clinical data |
| ☒ | ☐ Dual use research of concern |

## Methods

| n/a | Involved in the study |
|---|---|
| ☒ | ☐ ChIP-seq |
| ☒ | ☐ Flow cytometry |
| ☒ | ☐ MRI-based neuroimaging |

## Animals and other research organisms

Policy information about studies involving animals; ARRIVE guidelines recommended for reporting animal research, and Sex and Gender in Research.

| Laboratory animals | No laboratory animals were used for this study. |
|---|---|
| Wild animals | Only invertebrate animals (arthropods and earthworms) were collected and killed using ethanol during the study. This was necessary to assess biomass/density and community composition. Soil animals were extracted from soil samples using a Kempson extractor under a heat gradient of 40-45 grad Celsius above and 15 grad Celsius below the samples. Animals were first collected in dimethyleneglycol - water solution (1:1) and thereafter transferred into 70-80% ethanol solution. Understorey arthropods were collected using pan traps that stayed for 45 hours in the field. Arthropods where directly killed in the bowl that is filled with water and one drop of scentless soap. Each pan trap was shaken off through a sieve so that the arthropods could be collected in the sieve and then stored in a test tube containing 70% ethanol. The animals were collected by Anton Potapov with the research permit: 349/SIP/FRP/ES/Dit.KI/X/2016 (Validity 5 October 2016 to 5 October 2017) and 54/EXT/SIP/FRP/ES/Dit.KI/IX/2017 |
| Reporting on sex | (validity 04 October 2017 to 4 October 2018) and Isabelle Arimond with the research permit 370/SIP/FRP/ES/Dit. KI/X/2016 (Validity 20 October 2016 to 20 March 2017). Animal sex was not considered in the study. |
| Field-collected samples | After each round of fieldwork, samples were taken to the lab for identifications. Collected soil samples were transported in the lab within 2-3 days for heat extraction. All extracted invertebrates were stored in 70-80% ethanol solution and sorted to high-rank taxa under dissecting microscope. |
| Ethics oversight | No ethical approval was required for this study as it complies with regulations of the Collaborative Research Center 990 (project ID 192626868 – SFB 990) |

Note that full information on the approval of the study protocol must also be provided in the manuscript.

