## [Peer Review File · Nature]

Manuscript Title: Tree islands enhance biodiversity and functioning in oil palm landscapes

Reviewer Comments & Author Rebuttals

Reviewer Reports on the Initial Version:

Referee #1 (Remarks to the Author):

This study used a very nice field experiment in which 52 tree islands of different size and tree diversity were established in oil-palm background to show that such enrichment positively affects a large number of ecosystem services at landscape level without reducing oil-palm yield. This is an extremely interesting and important result with great relevance for application.

The experimental design is very well thought out with proper replication and randomization. I suggest that authors use their extended figure 11 instead of Fig. 1 to show this design. The number of measured ecosystem variables is very impressive, especially the near complete assessment of all groups of organisms potentially benefiting from the tree islands. Considering the wonderful design, I was disappointed by the simplistic data analysis and focus on SEMs, which are more a tool to discuss potential causal hypotheses than testing the effects for which the experiment was designed. With regard to the latter, I would have expected some linear model analysis with appropriate fixed- and random-effects terms. In the context of such a properly designed experiment it is quite clear how such models should be constructed and dangerous to use AICs or R²s to guide model selection.

Currently, instead of linear model analysis the authors use Kruskal-Wallis tests (with much less statistical power) and Chi-square statistics with nontransparent model assumptions and unclear statements about how hierarchical components of error were incorporated, e.g. regarding multiple measurements. The authors also mention they used transformations when data were not normally distributed. However, this requirement must be met by the residuals after a model has been fitted and not by the raw data.

One result demonstrates that large islands are better than small ones, but this statement does not consider that several small islands of the same total size as one large one could actually be better. This is not directly testable as a design factor, but still could be analyzed given the available data. I'm quite confident that several small will be more efficient than on large of the same size, even though a proper analysis would need a new experiment with a corresponding design.

Given the excellent design of the experiment, the authors should not be too shy to mention that this was done in a single model landscape and multiple landscapes should eventually be studied (see e.g. papers by Oehri et al. in PNAS 2017 and Nat.Comm. 2020).

With so many measurements it is difficult to present results in a comprehensive way and the authors do quite a good job regarding this issue. Nevertheless, the very many items of extended figures and tables make it hard to "plough" through all the details. I believe, here less would be more. As long as all the data are accessible on a repository, which I hope will be the case, interested readers can do

additional analyses themselves.

Overall, I really think this manuscript presents excellent work and want to stress that the above comments are only intended to strengthen the messages.

Referee #2 (Remarks to the Author):

The Zemp et al. manuscript describes positive outcomes for biodiversity and ecosystem function when oil palm plantations are enhanced with small islands of native trees. These results are original, important, and potentially high-impact. The manuscript is well-written and the references are appropriate. That said, I have concerns about some of the analyses and conclusions, as detailed below.

The topline results for the paper are that multidiversity increases 250% and multifunctionality increases 75% in plots on tree islands compared to plots in oil palm. I have two concerns about those topline results: one, I think the basis for measuring the increase in multidiversity and multifunctionality needs to be clarified, because I am increasing unclear on where those numbers come from, and two, the results showing a 250% increase in multidiversity and 75% increase in multifunctionality should be visualized in the main text. I don't think Figure 2 does that, which makes it a little misleading as the 'main' results figure in the manuscript. This is my understanding of the methods, using multidiversity as an example: per line 763, multidiversity was measured as the number of taxonomic groups (a maximum of 10) in which the number of species per plot exceeded 50% of the maximum number of species per plot in a particular habitat (tree island or oil palm). In other words, the 250% increase in multidiversity means that 2.5 times as many taxonomic groups had plot richness in tree islands exceeding 50% of the maximum richness in tree islands than taxonomic groups with plot richness in oil palm exceeding 50% of the number of species in oil palm. From Figure 2a & b, three taxonomic groups have greatest richness in oil palm, and seven have greatest richness on tree islands. $7 = 3 * 2.333$, so I initially thought that is where the 250% increase in multidiversity comes from. But then looking at Extended Data Table 6, I can see the value using the 50% threshold of 1.8 on tree islands and 0.5 in oil palm leads to a 250% increase in the value, so I then I figured that this is where the 250% increase comes from (I can also see the 75% increase in the Extended Data table for multifunctionality). But I don't know what these values are. I thought multidiversity was being measured as the number of taxonomic groups, but based on Figure 2 and Extended Data Table 6, that does not seem right. I don't understand what the 'average' and 'threshold' values are in Extended Data Table 6: certainly the average of all the numbers under 'Islands' in the first column of the table is not 0.03. So what is 0.03 an average of? This gets to the heart of my first concern, and the visualization for that result is the root of my second concern.

Line 118: Here you say that larger tree islands have enhanced per-area diversity, which reads to me as the number of species per ha or m² or some other standard unit of area. In the analysis I can see the effect of plot size. But plot size *is* the size of the tree island (line 328). You can't say that because there's more species in larger plots than in smaller plots there is enhanced per-area diversity. There is enhanced diversity in large plots compared to small plots probably mostly because of passive sampling. I don't have an issue with the analysis of increasing forest island size, but I don't

think this line should say there is enhanced *per-area* diversity, unless I missed a per-unit-area analysis somewhere.

Line 146: Continuing in the vein of the previous comment, I am not sure I completely 'see' the effect of tree island size on biodiversity. Looking at Extended Data Table 15, for the 10 diversity indicators, six increase with island size and 4 decrease. Only one of those changes is significant: the increase in tree species on tree islands, which of course increases because you sampled more individuals on large islands. Extended Data Table 10 (summary statistics for SEMs) shows (I believe) no differences in the SEM relationships between forest island size and any of the indicators. I know that 'effects' and 'statistical significance' are separate concepts. But on a fundamental level, I think that the result reported ('larger islands led to higher multidiversity') is not entirely supported by the data analysis. I suggest the authors either change the language about the effect of forest island size to be more circumspect, or demonstrate the reported result more robustly.

Line 508: How were the 30 reference palms selected? Was it a truly random selection? Haphazard? Something else? On the one hand this feels like a relatively minor point, but it's really not if we are to believe that oil palm production is not decreasing in landscapes with tree islands. So a bit more detail here is necessary.

Extended Data Figure 6: Based on this figure, it looks like overall yield was lowest in Year 4, the year the yield estimates used in the main text were calculated (line 518). Would the results concerning yield be qualitatively different if yield were calculated for other the years 1-3, using the yield of the third tree in the distance effect analysis as the reference (in addition to the trees in the control plots)? In a nutshell: is the non-significant difference in landscape yield described in the first paragraph an artefact of calculation during a low-yield year?

Also in Extended Data Figure 6: How are the 'total' values calculated? What are they totals of?

Extended Data Table 4: Why are the per area change values in the last row negative-shouldn't they be positive because the value on islands is greater than the number on the control plots?

Referee #3 (Remarks to the Author):

Dear authors,

You present a convincing case of biodiversity friendly intervention in a monocrop oil palm plantation enriched with "tree islands". While I might not call this ecological restoration (we are still far from the original vegetation that was cut to establish the plantation) this is a very interesting development. You show that enriching an oil palm plantation with patches of trees enhanced multiple species diversity by 250% and ecosystem functionality by 75% compared to the conventional monocultures. You also demonstrate that the larger tree islands led to higher multi-diversity and

multi-functionality gains and, of great interest for the managers that oil palm yields did not significantly decrease. Given the amount of land currently under oil palm monocultures and the expected development of the crop, this is indeed a very interesting result.

Your work is original and rather novel, as for a long time everyone was considering that the only way to industrial oil palm production was via monoculture with total eradication of prior vegetation and high intransit levels. This confirms also the recent body of literature about the ecological and economical benefits of oil palm agroforestry systems (Khasanh 2020, Susandi 2019, Mollins 2022).

Methods, experimental design, statistics, analyses and resulting conclusions are sound and robust. The duration of the experiment is already interesting but it would be fascinating to continue following the development of these tree islands to assess what will happen when the tree grow, compete; the structure complexify, etc.

Your discussion could be enriched by a quick overview of the other experiments on this oil palm agroforestry question. This would also add some more robustness and potential for extrapolation of your findings. If this was possible given the constraints in terms of manuscript length it would be a good thing.

Among references, I have been somewhat surprised not to find in this manuscript:

Zemp et al.(2019) Tree performance in a biodiversity enrichment experiment in an oil palm landscape. *Journal of Applied Ecology* 2019;56:23402-2352

Khasanah et al.(2020) Oil Palm Agroforestry Can Achieve Economic and Environmental Gains as Indicated by Multifunctional Land Equivalent Ratios. *Frontiers in Sustainable Food Systems*.
<https://doi.org/10.3389/fsufs.2019.00122>

Ahirwal et al. (2020) Oil palm agroforestry enhances crop yield and ecosystem carbon stock in northeast India: Implications for the United Nations sustainable development goals.
<https://doi.org/10.1016/j.spc.2021.12.022>

Author Rebuttals to Initial Comments:

Referee #1 (Remarks to the Author):

1. Comment:

This study used a very nice field experiment in which 52 tree islands of different size and tree diversity were established in oil-palm background to show that such enrichment positively affects a large number of ecosystem services at landscape level without reducing oil-palm yield. This is an extremely interesting and important result with great relevance for application.

Response: Thank you for the positive feedback.

2. Comment:

The experimental design is very well thought out with proper replication and randomization. I suggest that authors use their extended figure 11 instead of Fig. 1 to show this design.

Response: Following this suggestion, we have modified Figure 1 by including an improved version of Extended Figure 11 (showing the replication and randomization of our experiment), while keeping the details related to tree island area and diversity. We think that this new figure makes it easier for readers to understand the design of our experiment.

3. Comment:

The number of measured ecosystem variables is very impressive, especially the near complete assessment of all groups of organisms potentially benefiting from the tree islands.

Considering the wonderful design, I was disappointed by the simplistic data analysis and focus on SEMs, which are more a tool to discuss potential causal hypotheses than testing the effects for which the experiment was designed. With regard to the latter, I would have expected some linear model analysis with appropriate fixed- and random-effects terms. In the context of such a properly designed experiment it is quite clear how such models should be constructed and dangerous to use AICs or R²s to guide model selection.

Response: Following this suggestion, we added the results for individual linear models testing the influence of tree island area and tree diversity on each restoration outcome in Extended Data Figure 1 (model structure (1) below). For two reasons we opted to keep the piecewise structural equation modeling (SEM) results in the main text: i) the results from individual linear models are consistent with the piecewise SEMs and ii) our experiment was designed to test the potential benefits of tree islands on biodiversity and ecosystem functioning, as well as providing insights to changes in the mechanisms underlying biodiversity and ecosystem functioning across tree islands.

With the piecewise SEM we are able to test if the influence of tree island features (area and diversity) on biodiversity and ecosystem functioning indicators act primarily via tree dominance and structural complexity (i.e., indirect effects) or unmeasured mechanisms (i.e., direct effects).

Piecewise SEM uses a set of linear equations that are evaluated individually (Lefcheck 2016). Due to the fact that we performed separately piecewise models for each restoration outcome (one value per island), we used linear models instead of linear mixed effect models. In our case, our piecewise SEM include three linear models:

lm(restoration outcome ~ tree island area + tree diversity) (1)

lm(structural complexity ~ tree island diversity) (2)

lm(tree dominance ~ tree island area) (3)

Moreover, across each restoration outcome, experimental treatments were always included in linear model (1), with the model selection influencing only the inclusion of structural complexity and tree dominance as additional explanatory variables in this linear model. This is now clarified in the Method section (Lines 779 – 784 and 788).

4. Comment:

Currently, instead of linear model analysis the authors use Kruskal-Wallis tests (with much less statistical power) and Chi-square statistics with nontransparent model assumptions and unclear statements about how hierarchical components of error were incorporated, e.g. regarding multiple measurements.

Response: Thank you for this comment. Regarding the Kruskal-Wallis selection, we would like to clarify that we started the analysis by fitting a one-way Analysis of Variance (ANOVA) model. We tested the assumption of normally distributed residuals using the Shapiro-Wilk Normality Test and applied a suitable transformation of the explanatory variables if needed (with the Package bestNormalize version 1.6.1). As shown in Table 1 below, this methodological approach leads to similar significance statements as the Kruskal-Wallis tests in most cases. Yet, for many restoration outcomes, the distribution of the residuals remains non-normal even after correction. Therefore, we decided to use the Kruskal-Wallis test in the manuscript. Now, we have explicitly stated in the methods the reasons behind our decision (line 771). Furthermore, for transparency, we will provide the R code for analyzing the data with the ANOVA (including the tests of the assumptions and the appropriate correction when needed) and for comparison with the Kruskal-Wallis test for cases in which the assumptions for the ANOVA were met (see Code availability statement).

Regarding multiple measurements, we only used linear mixed effect models for per palm yield because we have multiple measurements within plots (Extended Data Fig. 5), which is not the case for the main restoration outcomes. As explained in the caption of Extended Data Fig. 5: “The tests are based on linear mixed-effect models with the annual yield of individual oil palms as the response variable and plot identity as the random effect. Multiple comparisons are conducted with a Tukey post-hoc test.” For more clarity, we now add the following sentence: “More details of the statistical analysis are provided in section 2.3.2 of Teuscher et al. 2017, where models number one, two, three and four correspond here to the panels a), b), c) and e), respectively.” (lines 1031 – 1035).

Table 1: Analysis of restoration outcomes (Biodiversity with species richness, Shannon and Simpson diversity, and ecosystem functioning) using Kruskal-Wallis and ANOVA. Agreement between statements of the significance (at 0.05) for the alternative analyses are presented for the restoration outcomes in which the assumptions for the ANOVA were met. (-) indicate models for which distribution of the residuals remain non-normal even after correction.

Restoration outcome	Kruskal-Wallis		ANOVA		Agreement
	Chi-squared	p-value	F	p-value	Yes / No
Species richness					
Bats	1.4053	0.2358	-	-	-
Birds	4.9902	0.0255	-	-	-
Arthropods	1.3497	0.2453	-	-	-
Herbs	0.8562	0.3548	0.7362	0.3947	Yes
Pollen	0.233	0.6293	0.0814	0.7766	Yes
Seeds	2.495	0.1142	14.0598	0.0039	No
Trees	9.3609	0.0022	-	-	-
Soil fauna	1.021	0.3123	-	-	-
Bacteria	0.2925	0.5886	-	-	-
Fungi	0.001	0.9746	0.1654	0.6859	Yes
Shannon diversity					
Bats	0.9804	0.3221	-	-	-
Birds	5.4477	0.0196	4.9216	0.0307	Yes
Arthropods	0.2925	0.5886	0.2439	0.6234	Yes
Herbs	1.7014	0.1921	1.2853	0.2619	Yes
Pollen	0.0425	0.8368	-	-	-
Seeds	5.247	0.022	-	-	-
Trees	9.3606	0.0022	-	-	-
Soil fauna	2.9514	0.0858	2.4448	0.1238	Yes
Bacteria	0.001	0.9746	-	-	-
Fungi	0.1457	0.7026	0.1991	0.6572	Yes

Simpson diversity

Bats	0.9804	0.3221	-	-	-
Birds	5.4472	0.0196	5.3558	0.0245	Yes
Arthropods	0.3279	0.5669	0.3848	0.5377	Yes
Herbs	1.2399	0.2655	-	-	-
Pollen	0.1427	0.7056	-	-	-
Seeds	4.6802	0.0305	-	-	-
Trees	9.3606	0.0022	-	-	-
Soil fauna	2.7368	0.0981	2.5749	0.1144	Yes
Bacteria	0.1711	0.6792	0.004	0.9496	Yes
Fungi	0.1012	0.7504	0.1795	0.6735	Yes

Ecosystem functioning

Oil palm yield	0.4899	0.484	-	-	-
Microclimate buffering	0.0041	0.9492	0.024	0.8775	Yes
Water infiltration	4.1457	0.0417	-	-	-
Evapotranspiration	1.5395	0.2147	-	-	-
Litter input	4.5435	0.033	-	-	-
Litter decomposition	0.4451	0.5047	-	-	-
Decomposers	0.3654	0.5455	0.3316	0.5671	Yes
Herbivores; soil fauna	0.4679	0.4939	-	-	-
Predators; soil fauna	3.0617	0.0802	-	-	-
Predators; arthropods	0.8832	0.3473	-	-	-
Predators; vertebrates	3.7662	0.0523	-	-	-
1 / Soil C:N	6.1579	0.0131	6.1073	0.0166	Yes
Soil decompaction	1.2424	0.265	-	-	-
Soil P	1.9595	0.1616	-	-	-
Pollination rate	0.004	0.9493	0.0004	0.9833	Yes
Pollinators	0.276	0.5994	-	-	-

Resistance to invasive plants	2.1516	0.1424	2.2874	0.1363	Yes
Native seeds	0.004	0.9493	-	-	-
Aboveground biomass	0.0648	0.7991	-	-	-
Tree growth	8.9801	0.0027	-	-	-

5. Comment:

The authors also mention they used transformations when data were not normally distributed. However, this requirement must be met by the residuals after a model has been fitted and not by the raw data.

Response: Thank you for the suggestion, we now use the untransformed data for all the analysis. We test the normality of the residuals after we fitted linear models and apply a suitable transformation of the explanatory variables to linearize the distribution of the residuals if needed. This is now also better explained in Methods (section "Structural equation modeling"): "We tested the assumption of normality of the residuals in models (1), (2) and (3) using the Shapiro -Wilk Normality Test, applied a suitable transformation of the explanatory variables if needed (Package bestNormalize version 1.6.1), and scaled all variables to zero mean and unit variance."(lines 794-797).

6. Comment:

One result demonstrates that large islands are better than small ones, but this statement does not consider that several small islands of the same total size as one large one could actually be better. This is not directly testable as a design factor, but still could be analyzed given the available data. I'm quite confident that several small will be more efficient than on large of the same size, even though a proper analysis would need a new experiment with a corresponding design.

Response: While the single large or several small (SLOSS) debate remains relevant for conservation in landscapes such as those in this study, we consider that doing so would extend considerably beyond the primary objective of our study, which is to assess the restoration outcomes of tree islands. Moreover, our sampling design in a single landscape, which uses a constant area within tree islands for most biodiversity and ecosystem functioning indicators, and therefore does not allow us to directly address the SLOSS debate. This would require comparing gamma diversity across multiple landscapes varying in number and sizes of patches (Fahrig et al. 2021; <https://doi.org/10.1111/brv.12792>), as well as sampling proportionally within habitat patches In the text, we now mention that "Understanding how biodiversity and ecosystem functioning changes in multiple landscapes ^{37,38} is urgently needed for designing and scaling up ecological restoration of oil palm landscapes worldwide." (lines 220 - 222).

7. Comment:

Given the excellent design of the experiment, the authors should not be too shy to mention that this was done in a single model landscape and multiple landscapes should eventually be studied (see e.g. papers by Oehri et al. in PNAS 2017 and Nat.Comm. 2020).

Response: Thank you Motivated by this comment, we added the following sentence to the conclusion: "While our study was conducted in a single landscape, it adds to other experimental^{31,34,35} and modeling evidence³⁶ of ecological and economic benefits in oil palm agroforestry systems. Understanding how biodiversity and ecosystem functioning changes in multiple landscapes^{37,38} is urgently needed for designing and scaling up ecological restoration of oil palm landscapes worldwide." (lines 218-222) and added the citation Oehri et al. Nat. Comm. 2020

8. Comment:

With so many measurements it is difficult to present results in a comprehensive way and the authors do quite a good job regarding this issue. Nevertheless, the very many items of extended figures and tables make it hard to "plough" through all the details. I believe, here less would be more. As long as all the data are accessible on a repository, which I hope will be the case, interested readers can do additional analyses themselves.

Response: In the revised version of this manuscript we have streamlined the presentation of our results by 1) systematically including rarefied tree diversity in all main analyses (instead of having this result as a separate figure); 2) showing the results of the linear models in a new figure (see Extended Fig. 1) that helps to visualize the results presented in the tables; 3) reducing the number of figures and tables in the supplementary information to the minimum needed for discussion in the main text; and; 4) moving all methodological details related to the calculation of per-palm and per-area yield to the supplementary information.

9. Comment:

Overall, I really think this manuscript presents excellent work and want to stress that the above comments are only intended to strengthen the messages.

Response: Thank you. Your constructive criticism helped us strengthen our manuscript.

Referee #2 (Remarks to the Author):

10. Comment:

The Zemp et al. manuscript describes positive outcomes for biodiversity and ecosystem function when oil palm plantations are enhanced with small islands of native trees. These results are original, important, and potentially high-impact. The manuscript is well-written and the references are appropriate. That said, I have concerns about some of the analyses and conclusions, as detailed below.

The topline results for the paper are that multidiversity increases 250% and multifunctionality increases 75% in plots on tree islands compared to plots in oil palm. I have two concerns about

those topline results: one, I think the basis for measuring the increase in multidiversity and multifunctionality needs to be clarified, because I am increasing unclear on where those numbers come from, and two, the results showing a 250% increase in multidiversity and 75% increase in multifunctionality should be visualized in the main text.

Response: Thank you very much for the positive feedback on our manuscript. Motivated by these comments, we now provide a clearer description of the results and how we communicate them in the Abstract. We have emphasized that, overall, seven of ten indicators of biodiversity increased on islands compared to oil palm plots and that 12 indicators of ecosystem functioning increased, five decreased and two remained neutral. Moreover, we now show that the number of indicators of biodiversity and ecosystem functioning that reach a certain threshold is systematically higher in the tree islands as compared to conventional oil palm plots, regardless of the threshold (see new Fig. 2d-f). For example, at 50% threshold, multidiversity increases by a factor of 1.5 in islands compared to oil palm, translating in four and 2.5 biodiversity indicators reaching at least 50% of their maximum observed species richness (i.e., within our landscape combining islands and conventional oil palm plots together), in tree islands and conventional oil palm plots, respectively. Similarly, six and three ecosystem functioning indicators reached at least 50% of their maximum observed values in tree islands and conventional oil palm plots, respectively. We updated the abstract (lines 75-78) and the main text (lines 141-146) accordingly.

11. Comment:

I don't think Figure 2 does that, which makes it a little misleading as the 'main' results figure in the manuscript. This is my understanding of the methods, using multidiversity as an example: per line 763, multidiversity was measured as the number of taxonomic groups (a maximum of 10) in which the number of species per plot exceeded 50% of the maximum number of species per plot in a particular habitat (tree island or oil palm). In other words, the 250% increase in multidiversity means that 2.5 times as many taxonomic groups had plot richness in tree islands exceeding 50% of the maximum richness in tree islands than taxonomic groups with plot richness in oil palm exceeding 50% of the number of species in oil palm.

Response: We would like to clarify that multidiversity represents the number of biodiversity indicators (i.e. species richness or Simpson diversity) that exceed a specified threshold, which is expressed as a percentage of the maximum observed values in the oil palm-dominated landscape. In other words, maximum diversity is calculated among all 56 study plots (based on both islands and oil palm plots together, and not in a particular habitat). This is now better explained in the caption of Fig. 2 (Lines 330-334), in the Methods (line 758) and in the main text (lines 141-146).

12. Comment:

From Figure 2a & b, three taxonomic groups have greatest richness in oil palm, and seven have greatest richness on tree islands. $7 = 3 * 2.333$, so I initially thought that is where the 250% increase in multidiversity comes from. But then looking at Extended Data Table 6, I can see the value using the 50% threshold of 1.8 on tree islands and 0.5 in oil palm leads to a 250% increase

in the value, so I then I figured that this is where the 250% increase comes from (I can also see the 75% increase in the Extended Data table for multifunctionality). But I don't know what these values are. I thought multidiversity was being measured as the number of taxonomic groups, but based on Figure 2 and Extended Data Table 6, that does not seem right. I don't understand what the 'average' and 'threshold' values are in Extended Data Table 6: certainly the average of all the numbers under 'Islands' in the first column of the table is not 0.03. So what is 0.03 an average of? This gets to the heart of my first concern, and the visualization for that result is the root of my second concern.

Response: Thank you for making this point. We improved the explanation in our Methods section and in the legend of the figures and Extended Data Tables the calculations behind multidiversity and multifunctionality values. Specifically, we used two approaches to calculate multidiversity and multifunctionality, average and threshold. For the average approach, we first standardized all individual biodiversity or ecosystem functioning indicators to unit scale and then calculated their average by plot. For the threshold approach, we calculated the number of indicators that exceed a specified threshold, with the threshold expressed as a percentage of the maximum observed values in our study landscape. For both approaches, we calculated median and mean values across either oil palm plots (i.e., control plots) and across islands (Extended Data Tables 4 and 6). The threshold approach for calculating multidiversity and multifunctionality is prominently presented in the main text (lines 141-146). In addition, for transparency, we now present changes across thresholds in multidiversity and multifunctionality in Fig. 2 (with new sub-panels d-f), including the values for each plot (see also response to comments 10 and 11). The threshold approach used here represents the state-of-the-art method in multifunctionality studies (Byrnes et al. 2014, Methods in Ecology and Evolution, Manning et al. 2018, Nature Ecology & Evolution)

13. Comment:

Line 118: Here you say that larger tree islands have enhanced per-area diversity, which reads to me as the number of species per ha or m² or some other standard unit of area. In the analysis I can see the effect of plot size. But plot size *is* the size of the tree island (line 328). You can't say that because there's more species in larger plots than in smaller plots there is enhanced per-area diversity. There is enhanced diversity in large plots compared to small plots probably mostly because of passive sampling. I don't have an issue with the analysis of increasing forest island size, but I don't think this line should say there is enhanced *per-area* diversity, unless I missed a per-unit-area analysis somewhere.

Response: We agree and have removed "per-area diversity". In our revised manuscript, we now write: "The influence of island area on biodiversity was not attributed to passive sampling²⁴, because all indicators have been standardized (by using constant sampling area or by using rarefaction curves, see Materials and Methods)" (lines 165-167). In the previous version, the only exception was tree diversity because the sampling area increases with plot size (i.e. all trees present in the plots were sampled) and therefore this variable was potentially affected by sampling effects. We now correct for this effect by estimating rarefied tree diversity and systematically use these estimates for all related analyses. This is now mentioned in the Materials and Methods (lines 369-282).

14. Comment:

Line 146: Continuing in the vein of the previous comment, I am not sure I completely 'see' the effect of tree island size on biodiversity. Looking at Extended Data Table 15, for the 10 diversity indicators, six increase with island size and 4 decrease. Only one of those changes is significant: the increase in tree species on tree islands, which of course increases because you sampled more individuals on large islands. Extended Data Table 10 (summary statistics for SEMs) shows (I believe) no differences in the SEM relationships between forest island size and any of the indicators. I know that 'effects' and 'statistical significance' are separate concepts. But on a fundamental level, I think that the result reported ('larger islands led to higher multidiversity') is not entirely supported by the data analysis. I suggest the authors either change the language about the effect of forest island size to be more circumspect, or demonstrate the reported result more robustly.

Response: In the revised version of this manuscript we improved the presentation of the effects of tree island size and planted tree diversity on the restoration outcomes (biodiversity and ecosystem functioning). Specifically, we show the effect sizes with (Fig. 3; Extended Data Figure 3) and without (Extended Data Fig. 1) considering the influence of vegetation structure (i.e. structural complexity and tree dominance). We show the results for each indicator separately, and aggregated for multidiversity and multifunctionality. These results clearly show that tree island area has a positive influence on many biodiversity indicators (e.g. trees, arthropods) - irrespectively of sampling effects (see also comment 13). We specify this more clearly in the main text (lines 161 - 165).

15. Comment:

Line 508: How were the 30 reference palms selected? Was it a truly random selection? Haphazard? Something else? On the one hand this feels like a relatively minor point, but it's really not if we are to believe that oil palm production is not decreasing in landscapes with tree islands. So a bit more detail here is necessary.

Response: We now specify in the main text that the 30 reference palms were selected "at approximately an equal distance to each tree island and whose neighborhood is typical for conventional oil palm monocultures" (lines 539 – 541). We also provide more details in the Supplementary note 1.

16. Comment:

Extended Data Figure 6: Based on this figure, it looks like overall yield was lowest in Year 4, the year the yield estimates used in the main text were calculated (line 518). Would the results concerning yield be qualitatively different if yield were calculated for other the years 1-3, using the yield of the third tree in the distance effect analysis as the reference (in addition to the trees in the control plots)? In a nutshell: is the non-significant difference in landscape yield described in the first paragraph an artefact of calculation during a low-yield year?

Response: Thank you for the feedback, which helps to improve the robustness of our results. The outcome of our experimental treatments are likely to change over time. Notably, the positive spill-over effects on the adjacent oil palms that we report in our study (November 2017 - October 2018, see Extended Data Figure 5) were already observed shortly after establishment of the experiment (in April 2015 - March 2016, see Teuscher et al. 2017). Furthermore, in order to assess how our results might change when using a different time period, we conducted the analysis for the entire duration of the field measurement campaign (November 2016 - October 2018). Per palm and per area oil yield for this period still indicates a beneficial effect of the experimental treatment on the adjacent oil palms (see Supplementary note Figure 4). As a consequence, the net change in oil palm yield within the tree islands (mean: 770.9 kg ± 1587 kg / island) remains higher than that of conventional oil palm monocultures (mean -58.4 ± 52 kg / plot) and the Kuskal-Wallis test indicates no significant differences ($p = 0.0858$). This suggests that our results are not an artifact of a particular year. We make this clear in the main text (lines 193 – 196) and provide the details in the Supplementary note 3.

17. Comment:

Also in Extended Data Figure 6: How are the 'total' values calculated? What are they totals of?

Response: The total refers to all 52 tree islands combined (diversity level 0 to 6). This is now clarified in the figure caption: "Relative changes in per area yield ($\Delta Y_{ha} / Y_{ha_ref}$ in %) compared to conventional oil palm monocultures ('reference') for different planted tree species diversity: 0, 1, 2, 3, 6 species, and all 52 tree islands combined ("total")" (line 1120).

18. Comment:

Extended Data Table 4: Why are the per area change values in the last row negative-shouldn't they be positive because the value on islands is greater than the number on the control plots?

Response: Yes, thank you for pointing out this mistake.

Referee #3 (Remarks to the Author):

19. Comment:

You present a convincing case of biodiversity friendly intervention in a monocrop oil palm plantation enriched with "tree islands". While I might not call this ecological restoration (we are still far from the original vegetation that was cut to establish the plantation) this is a very interesting development.

Response: Thank you for this comment. We could not agree more. We consider ecological restoration as a means "to safeguard biodiversity and ecosystem functioning while also promoting local livelihoods" (lines 95-96), which aligns with the definition of ecological restoration of the UN Decade as "the process of halting and reversing degradation, resulting in improved ecosystem

services and recovered biodiversity. Ecosystem restoration encompasses a wide continuum of practices, depending on local conditions and societal choice” (UNEP, 2021, p. 7). Yet, we emphasize in the abstract that these practices “should not replace the protection of remaining forests” (line 82).

20. Comment:

You show that enriching an oil palm plantation with patches of trees enhanced multiple species diversity by 250% and ecosystem functionality by 75% compared to the conventional monocultures. You also demonstrate that the larger tree islands led to higher multi-diversity and multi-functionality gains and, of great interest for the managers that oil palm yields did not significantly decrease. Given the amount of land currently under oil palm monocultures and the expected development of the crop, this is indeed a very interesting result.

Your work is original and rather novel, as for a long time everyone was considering that the only way to industrial oil palm production was via monoculture with total eradication of prior vegetation and high intransient levels. This confirms also the recent body of literature about the ecological and economical benefits of oil palm agroforestry systems (Khasanah 2020, Susandi 2019, Mollins 2022).

Response: Thank you. We included Khasanah et al. 2020 and other previous studies (see comment 22) as additional references.

21. Comment:

Methods, experimental design, statistics, analyses and resulting conclusions are sound and robust. The duration of the experiment is already interesting but it would be fascinating to continue following the development of these tree islands to assess what will happen when the trees grow, compete; the structure complexify, etc.

Response: We agree completely and we are working towards ensuring a long-term perspective.

22. Comment:

Your discussion could be enriched by a quick overview of the other experiments on this oil palm agroforestry question. This would also add some more robustness and potential for extrapolation of your findings. If this was possible given the constraints in terms of manuscript length it would be a good thing.

Among references, I have been somewhat surprised not to find in this manuscript:

Zemp et al.(2019) Tree performance in a biodiversity enrichment experiment in an oil palm landscape. Journal of Applied Ecology 2019;56:23402-2352

Khasanah et al.(2020) Oil Palm Agroforestry Can Achieve Economic and Environmental Gains as Indicated by Multifunctional Land Equivalent Ratios. Frontiers in Sustainable Food Systems. <https://doi.org/10.3389/fsufs.2019.00122>

Ahirwal et al. (2020) Oil palm agroforestry enhances crop yield and ecosystem carbon stock in northeast India: Implications for the United Nations sustainable development goals. <https://doi.org/10.1016/j.spc.2021.12.022>

Response: Thank you for the suggestions, we have integrated the three suggested references.

Reviewer Reports on the First Revision:

Referee #1 (Remarks to the Author):

I have reviewed this manuscript as Referee #1 in the first round and remain with my comments then that this is a very important manuscript with very valuable data but a very simplistic analysis. I'm not convinced by the excuses of the authors for keeping the simplistic analysis, but their clarifications make it easier to make concrete suggestions how a more appropriate analysis could be done. If the data would be available in a more convenient format than now, I would be glad to do some example analysis for the authors. Unfortunately, currently this would require major efforts because the data would have to be assembled from many sources (I tried).

Basically, the data should be organized in the following way: there are 56 plots, 4 controls and 52 islands. The latter follow a factorial island area x tree diversity design. For each plot there are multiple measurements, different years for tree growth, different taxa for richness other than trees, different variables for ecosystem functioning. The main hypothesis is the comparison between the 4 controls and the 52 islands (C) and the secondary hypotheses are potential effects of island area (A, log-linear [and factor if used as test for deviation from log-linearity]) and tree diversity (D, log-linear [and factor if used as test for deviation from log-linearity]). These hypotheses are tested against the variation among plots (P). The multiple measurements (say T for the 10 taxa shown in Fig. 2a, b [including possible contrasts such as trophic levels]) are like split-plot treatments within plots. Thus, a general model for analysis, using R's `lm()` notation, could be:

$$y \sim C / (A * D) + P + T + T:(C / (A * D)),$$

where "`C / (A * D)`" can also be written as "`C + A + D + A:D`" if sequentially fitted in this order. Note that in this case A and D should have a unique level for control plots, e.g. 99, and that in this case C must be fitted before A and D. If `lm()` is used, the terms before P have to be tested against P (and you have to use `keep.order=T`) and if `lme()` is used P is the random term (as the authors did for tree growth).

Instead of the above, the authors test C separately, still using non-parametric statistics and Chi2 tests. They also tried "automatic" transformations (and in the rebuttal say they did this for explanatory terms whereas errors in linear models only play a role in dependent variables) and let themselves be guided by normality of residuals tests. Both is not very transparent and risky in cases where group sizes are so different ($n = 4$ for control plots) and furthermore it is well known that the requirement for normally distributed residuals is not so strict in linear modelling. More importantly, by doing the test of C separately, the authors (i) miss the opportunity to use the better residuals from the the model "`C / (A * D)`", which may well be more normal than the residuals without `A * D` (as the authors currently do) and (ii) use the 52 islands in two separate analysis, ignoring the inter-dependence of the two.

Instead of including the multiple measurements of taxa richness and ecosystem functioning in a single model (as the authors correctly do for the multiple measurements of oil-palm variables over the five years, albeit without separating linear year and factor year), the authors do multiple

separate analyses, again not considering their inter-dependence and adjustments for multiple testing. Instead they use simple counts of positive and negative responses of measurements in one-way analyses for each explanatory term (e.g. A and D) separately and multidiversity and multifunctionality constructs to integrate multiple measurements. Again, these latter are derived in a way that I don't find very transparent.

The only place where the authors do include more than one explanatory term in analysis is in their SEMs, but even here only as $y \sim A + D$, and as far as I understand, but this is not clearly explained, excluding control plots (C). Furthermore, it seems they standardized A and D for the SEMs, which makes the corresponding $\text{Im}(y \sim A + D)$ very different from the one without standardization. I recommend that the authors do not standardize the variables but rather keep them as in the linear models (e.g. $A = \log(\text{area})$ and $D = \log(\text{diversity})$) as the output will still provide the same standardized solutions for path coefficients.

So, my overall recommendation is that the authors more clearly formulate their hypotheses about C, A and D effects and then test them together in a single model. Regarding the multiple measurements I would recommend to include them as well (see above), or, as a simpler alternative, they could calculate effect sizes using $\text{Im}(y \sim C / (A * D))$ for all measurements and subsequently use meta-analysis with those effect sizes (rather than using simple counts of positive and negative effect sizes from $\text{Im}(y \sim A)$ and $\text{Im}(y \sim D)$). Once the new analyses have been done, I'll be happy to look at this manuscript again and comment on some smaller issues I noted for myself.

Referee #2 (Remarks to the Author):

I think the authors have done a nice job with their response, and think the paper is ready to be accepted.

Referee #3 (Remarks to the Author):

Dear authors

Many thanks for addressing the concerns of the reviewers. Your manuscript is stronger following the new and revised analyses. This will be an important contribution to the literature and hopefully to the design of new palm oil plantations (likely valid for any monoculture I believe).

Looking forward to see it published and possibly some more analyses in a few years from now.

Best regards

Author Rebuttals to First Revision:

Referee #1 (Remarks to the Author):

I have reviewed this manuscript as Referee #1 in the first round and remain with my comments then that this is a very important manuscript with very valuable data but a very simplistic analysis. I'm not convinced by the excuses of the authors for keeping the simplistic analysis, but their clarifications make it easier to make concrete suggestions how a more appropriate analysis could be done. If the data would be available in a more convenient format than now, I would be glad to do some example analysis for the authors. Unfortunately, currently this would require major efforts because the data would have to be assembled from many sources (I tried).

Basically, the data should be organized in the following way: there are 56 plots, 4 controls and 52 islands. The latter follow a factorial island area x tree diversity design.

Response: We are grateful for the time and effort invested in reviewing our manuscript and for the constructive suggestions to improve our analyses. We have now tested and implemented the suggested analysis. Furthermore, we have made data, in a convenient format, and R code publicly available: <https://github.com/Clara-Zemp/efforts-bee-synthesis.git>.

The GitHub' repository contains all individual and assembled data (as original and standardized values), for biodiversity:

- *AllBioData_LongFormat170123.csv*
- *AllBioData_StandardizedLongFormat310123.csv*

and ecosystem functioning:

- *AllFunctionsData_LongFormat170123.csv*
- *AllFuncionsDataStandadized_LongFormat250123.csv*.

Upon acceptance, the R code and data will also be placed on Zenodo.

For each plot there are multiple measurements, different years for tree growth, different taxa for richness other than trees, different variables for ecosystem functioning.

Response: All variables presented in our paper were measured during a limited time period (from October 2016 to October 2018, including tree growth, see Extended Data Tables 1 and 2). The only exception was "Per-Palm Yield" which has been continuously monitored since the establishment of the experiment. However, this variable was not part of the 19 ecosystem functioning indicators presented in the main text. It was instead presented in the additional information (Extended Data Figure 4). In the revised manuscript, we mentioned: "Oil palm yield was continuously measured since the beginning of the experiment at the level of individual palm, but the data was then aggregated over space and time (see below). Each variable presented in the main text had one measurement per plot." (Line 421-423)

The main hypothesis is the comparison between the 4 controls and the 52 islands (C) and the secondary hypotheses are potential effects of island area (A, log-linear [and factor if used as test for deviation from log-linearity]) and tree diversity (D, log-linear [and factor if used as test for deviation from log-linearity]). These hypotheses are tested against the variation among plots (P). The multiple measurements (say T for the 10 taxa shown in Fig. 2a, b [including possible contrasts such as trophic levels]) are like split-plot treatments within plots. Thus, a general model for analysis, using R's `lm()` notation, could be:

$$y \sim C / (A * D) + P + T + T:(C / (A * D)),$$

where "C / (A * D)" can also be written as "C + A + D + A:D" if sequentially fitted in this order. Note that in this case A and D should have a unique level for control plots, e.g. 99, and that in this case C must be fitted before A and D. If `lm()` is used, the terms before P have to be tested against P (and you have to use `keep.order=T`) and if `lme()` is used P is the random term (as the authors did for tree growth).

Response: We thank the reviewer for the thoughtful suggestion. To implement the suggested model(s), for the controls, we set $A = 10$ and $D = 0$ (no planted tree species). In addition, A and D were log-transformed.

First, we fitted the suggested linear models using `lm()` and `keep.order = T`. However, this argument is no longer supported in the current version of 'lm' or 'aov' (warning message: "In `lm.fit(x, y, offset = offset, singular.ok = singular.ok, ...)`: extra argument 'keep.order' will be disregarded"). We therefore decided to fit the suggested linear mixed effect models, a robust analytical way for analyzing biodiversity experiments (Schmid et al. 2017).

Schmid, Bernhard, Martin Baruffol, Zhiheng Wang, and Pascal A. Niklaus. "A Guide to Analyzing Biodiversity Experiments." *Journal of Plant Ecology* 10, no. 1 (2017): 91. <https://doi.org/10.1093/jpe/rtw107>.

The model structure was $y \sim C / (A * D) + T + T:(C / (A * D))$ as fixed effects and included P as the random effect.

All biodiversity metrics (species richness, Shannon diversity and Simpson diversity) were in the same units, i.e. effective number of species. However, within each of our diversity metrics, there was high variability across biodiversity indicators (Response letter Fig. 1), which influenced model fit. Ecosystem functions exhibited even greater variation (Response letter Fig 2) due to the differences in measurement units, with for instance, oil palm yield varying between -3788 and +5703 in kg / island (net change) and microclimate buffer between 0.12 and 0.18 °C (daily amplitude of air temperature). Therefore, we standardized the data to unit scale (using the min and max values) before fitting the respective linear mixed effect models and this improved the residual distributions and diagnosis (Response letter Fig. 3).

Results of the integrated models are presented in the table below (see also Extended Data Tables 4 and 5 of the revised manuscript).

Fig. 1: Histogram of the biodiversity data before (upper panel) and after (bottom panel) standardization to unit scale.

Fig. 2: Histogram of the ecosystem functioning data before (up) and after (bottom) standardization to unit scale

Fig 3. Residual diagnostics for the linear mixed effect model using DHARMA for Species richness without (left) and with (right) standardization to unit scale. Similar improvements were observed for Shannon and Simpson diversity and for ecosystem functioning when standardizing data prior to model fit.

Table 1: ANOVA of the linear mixed effect models, for biodiversity and for ecosystem functions using the standardized values.

Species richness

	numDF	denDF	F-value	p-value
(Intercept)	1	459	5437.892	<.0001
Treatment	1	51	10.793	0.0018
SizeLog	1	51	3.793	0.0570
DivLevLog	1	51	0.600	0.4423
Taxa	9	459	37.994	<.0001
SizeLog:DivLevLog	1	51	0.201	0.6554
Treatment:Taxa	9	459	2.545	0.0074
SizeLog:Taxa	9	459	5.059	<.0001
DivLevLog:Taxa	9	459	1.310	0.2291
SizeLog:DivLevLog:Taxa	9	459	1.453	0.1628

Shannon diversity

	numDF	denDF	F-value	p-value
(Intercept)	1	459	3519.813	<.0001
Treatment	1	51	4.579	0.0372
SizeLog	1	51	5.798	0.0197
DivLevLog	1	51	1.976	0.1658
Taxa	9	459	42.321	<.0001
SizeLog:DivLevLog	1	51	1.465	0.2318
Treatment:Taxa	9	459	3.589	0.0002
SizeLog:Taxa	9	459	2.805	0.0033
DivLevLog:Taxa	9	459	2.326	0.0144
SizeLog:DivLevLog:Taxa	9	459	1.380	0.1945

Simpson diversity

	numDF	denDF	F-value	p-value
(Intercept)	1	459	2887.7879	<.0001
Treatment	1	51	4.7155	0.0346
SizeLog	1	51	3.9752	0.0515
DivLevLog	1	51	3.7295	0.0590
Taxa	9	459	24.5455	<.0001
SizeLog:DivLevLog	1	51	0.8520	0.3603
Treatment:Taxa	9	459	3.0471	0.0015
SizeLog:Taxa	9	459	1.8360	0.0598
DivLevLog:Taxa	9	459	2.7682	0.0037
SizeLog:DivLevLog:Taxa	9	459	1.1635	0.3166

Ecosystem functioning

	numDF	denDF	F-value	p-value
(Intercept)	1	918	4233.882	<.0001
Treatment	1	51	6.233	0.0158
SizeLog	1	51	12.851	0.0008
DivLevLog	1	51	0.216	0.6438
Function	18	918	95.173	<.0001
SizeLog:DivLevLog	1	51	2.194	0.1447
Treatment:Function	18	918	0.918	0.5564
SizeLog:Function	18	918	1.382	0.1319
DivLevLog:Function	18	918	0.987	0.4718
SizeLog:DivLevLog:Function	18	918	0.930	0.5410

Instead of the above, the authors test C separately, still using non-parametric statistics and Chi2 tests. They also tried "automatic" transformations (and in the rebuttal say they did this for explanatory terms whereas errors in linear models only play a role in dependent variables) and let themselves be guided by normality of residuals tests. Both is not very transparent and risky in cases where group sizes are so different (n = 4 for control plots) and furthermore it is well known that the requirement for normally distributed residuals is not so strict in linear modelling.

More importantly, by doing the test of C separately, the authors (i) miss the opportunity to use the better residuals from the the model "C / (A * D)", which may well be more normal than the residuals without A * D (as the authors currently do) and (ii) use the 52 islands in two separate analysis, ignoring the inter-dependence of the two.

Instead of including the multiple measurements of taxa richness and ecosystem functioning in a single model (as the authors correctly do for the multiple measurements of oil-palm variables over the five years, albeit without separating linear year and factor year), the authors do multiple separate analyses, again not considering their inter-dependence and adjustments for multiple testing.

Response: Thank you for these comments. In the revised version of the manuscript, we now show the results for the linear mixed effects models integrating multiple biodiversity or ecosystem functioning measurements (Extended Data Tables 4 and 5). For the linear mixed effect models, we tested the effects of C, A and D jointly, instead of testing them separately (Line 812 - 825). As mentioned, this allowed for considering the interdependence and adjustments for multiple testing, and for including interactions. While we did not apply transformation on the data, we systematically standardized the data prior to fitting the linear mixed effects models for the above-mentioned reasons.

The results from the linear mixed effect models supported our previous findings, i.e. that :

- *“tree islands had higher biodiversity and ecosystem functioning compared to conventionally managed oil palm monocultures” (lines 133-134)*

- “larger islands results in restoration benefits, both in terms of ecosystem functioning and biodiversity” (Lines 163-164)
- “The effect of planted tree diversity [...] depended on the biodiversity indicator” (Line 188)

Instead they use simple counts of positive and negative responses of measurements in one-way analyses for each explanatory term (e.g. A and D) separately and multidiversity and multifunctionality constructs to integrate multiple measurements. Again, these latter are derived in a way that I don't find very transparent.

Response: We apologize for the lack of clarity regarding the responses of multidiversity and multifunctionality in the previous version of the manuscript, which have now been addressed. Previously, we indeed described our results using the number of positive, negative or neutral responses. Yet, neither multidiversity or multifunctionality indices are based on the counts of positive or negative responses, but rather on the average level of biodiversity or ecosystem functioning indicators that surpass a certain threshold (Manning et al. 2018; Grass et al. 2020). In the revised version of the manuscript, we removed from the Abstract the sentence “Seven of ten indicators of biodiversity increased and 12 indicators of ecosystem functioning increased, five decreased and two did not change on islands compared to oil palm plots, resulting in”. Instead, we state that “Overall, indicators of biodiversity and ecosystem functioning, as well as multidiversity and ecosystem multifunctionality, were higher in tree islands compared to conventionally managed oil palm” (lines 75-76). Our main results are now backed up by the new statistical analyses. Yet, we kept some of the descriptions for single biodiversity and functions indicators when describing the results in the main text to illustrate the magnitude of the change either in number of species or in percentage, metrics that we think are readily understood by a broad audience.

Multidiversity and multifunctionality provide a holistic overview of measured biodiversity and ecosystem functioning that is not possible by presenting variables individually. The method for calculation based on thresholds has the advantage of better considering trade-offs and synergies between the variables than alternative approaches like linear modeling. The method for calculation is well described and discussed in the following references.

This is now better explained in the main text:

“To provide a holistic overview of biodiversity and ecosystem functioning across the experiment, we calculated multidiversity and multifunctionality using the aforementioned indicators (Manning et al. 2018)”. (Lines 120-121)

“The calculation of multidiversity (or multifunctionality) relies on the number of biodiversity (or functioning) indicators that cross a certain threshold (Manning et al. 2018), with thresholds expressed as the percentage of the maximum observed values within the study system (here, within our landscape combining islands and conventionally managed oil palm monocultures).” (Line 147 - 151).

Relevant references for multifunctionality:

- Hector, A., & Bagchi, R. (2007). Biodiversity and ecosystem multifunctionality. *Nature*, 448(7150), 188-190.
- Byrnes, J. E., Gamfeldt, L., Isbell, F., Lefcheck, J. S., Griffin, J. N., Hector, A., ... & Emmett Duffy, J. (2014). Investigating the relationship between biodiversity and ecosystem multifunctionality: challenges and solutions. *Methods in Ecology and Evolution*, 5(2), 111-124.

- Manning, P., Van Der Plas, F., Soliveres, S., Allan, E., Maestre, F. T., Mace, G., ... & Fischer, M. (2018). Redefining ecosystem multifunctionality. *Nature ecology & evolution*, 2(3), 427-436.

and multidiversity:

- Martin, D. A., Andrianisaina, F., Fulgence, T. R., Osen, K., Rakotomalala, A. A. N. A., Raveloaritiana, E., ... & Kreft, H. (2022). Land-use trajectories for sustainable land system transformations: Identifying leverage points in a global biodiversity hotspot. *Proceedings of the National Academy of Sciences*, 119(7), e2107747119.
- Grass, I., Kubitzka, C., Krishna, V. V., Corre, M. D., Mußhoff, O., Pütz, P., ... & Wollni, M. (2020). Trade-offs between multifunctionality and profit in tropical smallholder landscapes. *Nature communications*, 11(1), 1186.

The only place where the authors do include more than one explanatory term in analysis is in their SEMs, but even here only as $y \sim A + D$, and as far as I understand, but this is not clearly explained, excluding control plots (C).

Response: We use the SEMs to provide a mechanistic understanding of how planted tree diversity and island size affect diversity and ecosystem functioning **within** tree islands. In other words, the SEMs reveal if the effect of tree islands on biodiversity and ecosystem functioning is mediated by planted tree diversity or island size (either indirectly via changes in vegetation structure or through alternative mechanisms, i.e., direct effects). We have clarified this in the revised version of the manuscript: "To provide a mechanistic understanding of the effects of planted tree diversity and island area on biodiversity and ecosystem functioning, we also measured 12 indicators of vegetation structure (Extended Data Table 3)" (lines 120-122).

Because our main aim with the SEMs was to understand the variation within the 52 tree islands as a result of our treatments and not the control plots, which have a fixed area and are managed conventionally (i.e. no oil palm was felled, no tree was planted and application of fertilizer, herbicide and pesticide was as usual - Line 408 - 209), we decided to exclude them for this particular analysis.

Furthermore, it seems they standardized A and D for the SEMs, which makes the corresponding $\ln(y \sim A + D)$ very different from the one without standardization. I recommend that the authors do not standardize the variables but rather keep them as in the linear models (e.g. $A = \log(\text{area})$ and $D = \log(\text{diversity})$) as the output will still provide the same standardized solutions for path coefficients.

Response: Thank you for the recommendation. In the SEM, A and D were log-transformed (e.g. $A = \log(\text{area})$ and $D = \log(\text{diversity} + 1)$) but not standardized, as in the linear mixed-effects models. Furthermore, following the suggestion above, we no longer standardized the response variable prior to fitting the SEMs but applied transformations when these were necessary to improve model fits (Response letter Figure 4).

For more transparency, we now clearly mention that "The transformation concerned four out of ten indicators for species richness and Shannon diversity, three indicators out of ten indicators for Simpson diversity and 14 indicators out of 19 for ecosystem functioning." (Line 854-856). Details about the transformation can be obtained from the code available in Github (file: 06-SEM.R).

In the revised version of the manuscript, we now show the standardized path coefficients.

Posterior Predictive Check

Model-predicted lines should resemble observed data line

Linearity

Reference line should be flat and horizontal

Homogeneity of Variance

Reference line should be flat and horizontal

Influential Observations

Points should be inside the contour lines

Collinearity

High collinearity (VIF) may inflate parameter uncertainty

Normality of Residuals

Dots should fall along the line

Fig. 4: One example of problematic SEM fit when using untransformed data: Density of native seeds as a function of planted tree diversity (divlev - log-transformed), island area (plotsize - log-transformed). In such cases, we transformed the response variable to improve the SEM model fit. Model residual diagnostic was performed with the “performance” R package (Lüdtke et al., 2021).

So, my overall recommendation is that the authors more clearly formulate their hypotheses about C, A and D effects and then test them together in a single model.

Regarding the multiple measurements I would recommend to include them as well (see above), or, as a simpler alternative, they could calculate effect sizes using $\text{lm}(y \sim C / (A * D))$ for all measurements and subsequently use meta-analysis with those effect sizes (rather than using simple counts of positive and negative effect sizes from $\text{lm}(y \sim A)$ and $\text{lm}(y \sim D)$). Once the new analyses have been done, I'll be happy to look at this manuscript again and comment on some smaller issues I noted for myself.

Response: We thank the reviewer for their constructive suggestions. We now formulate our hypotheses more clearly, for example: "Planted diversity effects on restoration outcomes are likely mediated by higher vegetation structural complexity, i.e. the three-dimensional distribution of plants within an ecosystem²¹." (Line 128-130)

We have integrated the constructive feedback into the revised version of this manuscript, which has contributed considerably to strengthening it.

Referee #2 (Remarks to the Author):

I think the authors have done a nice job with their response, and think the paper is ready to be accepted.

Referee #3 (Remarks to the Author):

Dear authors

Many thanks for addressing the concerns of the reviewers. Your manuscript is stronger following the new and revised analyses. This will be an important contribution to the literature and hopefully to the design of new palm oil plantations (likely valid for any monoculture I believe).

Looking forward to see it published and possibly some more analyses in a few years from now.

Response: We thank you both for the positive comments and for the time and effort invested in reviewing the previous version of our manuscript.

Reviewer Reports on the Second Revision:

Referee #1 (Remarks to the Author):

This time the authors implemented the suggested linear-mixed model to test their hypotheses. I checked both the authors responses to the review comments and the implementation of these into the revised text and am now very happy with the outcome. I have no further comments. All the statistic analyses are now appropriate. This will be a very important contribution to our knowledge about biodiversity benefits in agri- or silvicultural landscapes.

Author Rebuttals to Second Revision:

Referees' comments:

Referee #1 (Remarks to the Author):

This time the authors implemented the suggested linear-mixed model to test their hypotheses. I checked both the authors responses to the review comments and the implementation of these into the revised text and am now very happy with the outcome. I have no further comments. All the statistic analyses are now appropriate. This will be a very important contribution to our knowledge about biodiversity benefits in agri- or silvicultural landscapes.

Response: We would like to thank you once again for your time and effort in reviewing our manuscript, and for the concrete suggestions about the linear mixed model. We are also happy with the implemented changes and hope that our manuscript will contribute to advance knowledge needed to enhance biodiversity and ecosystem functioning in oil palm landscapes.

Following the editorial requests, we have deleted the figure about oil palm yield over a longer time period (previously referred to as Supplementary Note Figure 4, see below). The main results previously included in the figure caption have been added in the Supplementary Note 1.

In addition, we have added a figure describing the clustering analysis of ecosystem functioning indicators, which improves the description of the method (see Supplementary Note 4).

We have implemented all other editorial requests. The modifications to the manuscript are highlighted in colored text (blue).

Supplementary Note Figure 4 | Effects of the experimental treatment on per palm and per area yield. Each dot represents an individual oil palm that was monitored within our study (N=404 in total, from October 2016 to November 2018). In each boxplot, the horizontal line corresponds to the median; the lower and upper hinges correspond to the first and third quartiles and the upper and lower whiskers are limited by the 1.5 the interquartile ranges. For direct comparison with earlier findings, here we followed established methodology²⁷. The tests are based on linear mixed-effect models with the annual yield of individual oil palms as the response variable and the plot identity as the random effect. Multi-comparison is conducted with a post-hoc Tukey test. More details of the statistical analysis are provided in section 2.3.2 of Teuscher et al. 2017, where models number one, two, three and four correspond here to the panels a), b), c) and e), respectively. The overall effects of the experimental treatment on **(a)** per palm yield (in kg/palm) and **(b)** per area yield (in kg/ha) were significant in both cases ($\chi^2 = 19.61$, $df = 2$, p -value < 0.001 and $\chi^2 = 132.71$, $df = 2$, p -value < 0.001 , respectively). Multi-comparison indicates that **(a)** per palm yield inside the islands was lower than per palm yield adjacent to the tree islands and per palm yield in the conventional oil palm; **(b)** per area yield inside the tree islands were lower than per area yield adjacent to the tree islands and per area yield in the conventional oil palm. **(c)** The distance to the tree island edge (i.e. positions 1, 2 and 3) had an effect on the per palm yield of the adjacent oil palms ($\chi^2 = 9.36$, $df = 2$, p -value = 0.009). **(d)** There was no effect of tree planting on the per palm yield of the adjacent oil palms ($\chi^2 = 1.54$, $df = 1$, p -value = 0.21). **(e)** The effect of oil palm thinning on per palm yield of the adjacent palms was marginally significant ($\chi^2 = 3.42$, $df = 1$, p -value = 0.06). Significance levels are indicated by . ($p < 0.1$), * ($p < 0.05$), ** ($p < 0.01$) and *** ($p < 0.001$).